# Beyond Self-Repellent Kernels: History-Driven Target Towards Efficient Nonlinear MCMC on General Graphs

Jie Hu [1]   Yi-Ting Ma [1]   Do Young Eun [1]

## Abstract

We propose a *history-driven target (HDT)* framework in Markov Chain Monte Carlo (MCMC) to improve any random walk algorithm on discrete state spaces, such as general undirected graphs, for efficient sampling from target distribution $\mu$. With broad applications in network science and distributed optimization, recent innovations like the self-repellent random walk (SRRW) achieve near-zero variance by prioritizing undersampled states through transition kernel modifications based on past visit frequencies. However, SRRW's reliance on explicit computation of transition probabilities for all neighbors at each step introduces substantial computational overhead, while its strict dependence on time-reversible Markov chains excludes advanced non-reversible MCMC methods. To overcome these limitations, instead of direct modification of transition kernel, HDT introduces a history-dependent target distribution $\pi[\mathbf{x}]$ to replace the original target $\mu$ in any graph sampler, where $\mathbf{x}$ represents the empirical measure of past visits. This design preserves lightweight implementation by requiring only local information between the current and proposed states and achieves compatibility with both reversible and non-reversible MCMC samplers, while retaining unbiased samples with target distribution $\mu$ and near-zero variance performance. Extensive experiments in graph sampling demonstrate consistent performance gains, and a memory-efficient Least Recently Used (LRU) cache ensures scalability to large general graphs.

[1]Department of Electrical and Computer Engineering, North Carolina State University, Raleigh, North Carolina, USA. Correspondence to: Do Young Eun <dyeun@ncsu.edu>.

*Proceedings of the $42^{nd}$ International Conference on Machine Learning*, Vancouver, Canada. PMLR 267, 2025. Copyright 2025 by the author(s).

## 1. Introduction

Random walks on general graphs are fundamental tools across diverse disciplines, including physics, statistics, machine learning, biology, and social sciences (Robert & Casella, 2013; Grover & Leskovec, 2016; Masuda et al., 2017; Kim et al., 2024), powering applications like online social networks (Xie et al., 2021), web crawling (Olston et al., 2010), robotic exploration (Placed et al., 2023), and mobile networks (Triastcyn et al., 2022; Ayache et al., 2023). Starting from any state (node), a walker transitions to one of its neighbors along the connected edges using only local information, combining simplicity with scalability. This makes random walks powerful for exploring large networks and performing graph-based Markov chain Monte Carlo (MCMC) tasks to generate samples from target distributions $\mu$ on the graph (e.g., uniform, degree-based, or energy-based). They also enable distributed optimization in networked systems (Sun et al., 2018; Hu et al., 2022; Even, 2023; Hendrikx, 2023). Simple random walks (Gjoka et al., 2011; Perozzi et al., 2014) and Metropolis-Hastings (MH) random walks (Metropolis et al., 1953; Hastings, 1970; Xia et al., 2019) are foundational methods for these tasks, though they can suffer from slow mixing and local trapping.

### 1.1. Recent Advances in Graph Sampling

In social networks, e-commerce, recommendation systems and other domains, practitioners need to estimate graph size, node degree distribution, and label distribution (Xie et al., 2021). These applications include detecting bot populations, identifying high-value user segments or scarce features (Nakajima & Shudo, 2023), and gradient of local loss for token algorithms in distributed optimizations (Sun et al., 2018; Even, 2023), especially with limited graph access. However, a standard random walk can be slow in discovering underrepresented regions or rare features, forcing more samples to achieve acceptable levels of accuracy. To improve the sampling efficiency of the MH algorithm, MH with Delayed Rejection reduces persistent rejections to facilitate exploration (Green & Mira, 2001). Another variant is the Multiple-Try Metropolis (MTM), which proposes multiple candidates at each step and selects one based on their weights (Liu et al., 2000; Pandolfi et al., 2010).

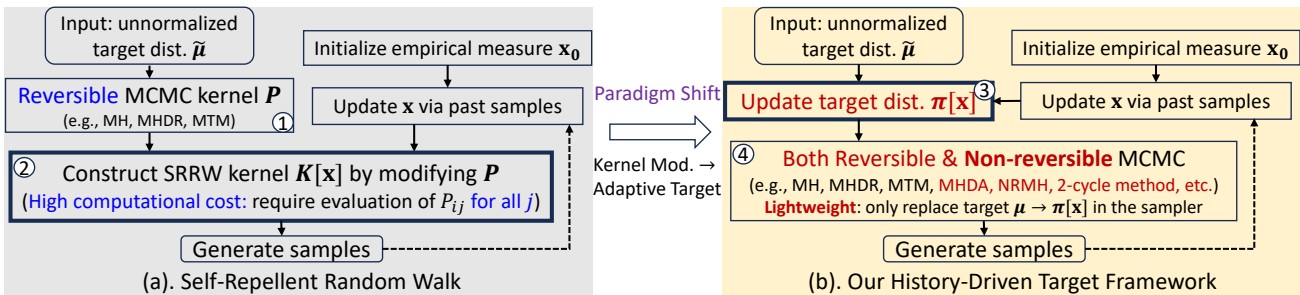

Figure 1. Flowchart comparison between (a) SRRW (Doshi et al., 2023) and (b) our History-Driven Target (HDT) framework.

The choice of weights is further refined by Chang et al. (2022) with the integration of locally balanced functions in Zanella (2020), quantifying the improvement in mixing time. Further improvements have been achieved via non-reversible random walk techniques. These include lifted Markov chains, which augment the state space with directional information (Chen et al., 1999; Diaconis et al., 2000; Apers et al., 2017); 2-cycle MCMC methods, which alternate between two reversible chains (Maire et al., 2014; Ma et al., 2016; Andrieu & Livingstone, 2021); and modifications from a reversible chain by adding antisymmetric perturbations (Suwa & Todo, 2010; Chen & Hwang, 2013; Bierkens, 2016; Thin et al., 2020). Non-backtracking random walks, which avoid revisiting the immediate past, have also improved exploration (Alon et al., 2007; Lee et al., 2012; Hermon, 2019).

In fields such as statistical physics (e.g., Ising or Potts models (Grathwohl et al., 2021; Zhang et al., 2022)), machine learning (e.g., energy-based models for image tasks or Bayesian methods for variable selection (Hinton, 2012; Sun et al., 2023a)), and econometrics (e.g., MCMC-MLE for random graph models (Bhamidi et al., 2011; Byshkin et al., 2016)), a discrete state space can be represented as a graph. Here, nodes denote configurations, which are linked by edges via predefined Hamming distance. Efficient exploration yields quicker, more accurate system representation and inference. Specifically, informed proposals (Zanella, 2020) utilize local data to optimize acceptance-exploration balance but struggle with large-scale applicability. Gradient-based methods improve these proposals through techniques like Taylor series, Metropolis-adjusted Langevin, and Newton approximations (Grathwohl et al., 2021; Rhodes & Gutmann, 2022; Zhang et al., 2022; Sun et al., 2023b; Xiang et al., 2023). Nonetheless, they are designed for structured spaces and specific energy functions, unlike our focus on general graphs and arbitrary distributions $\boldsymbol{\mu}$, which renders gradient-based MCMC methods inapplicable.

All the aforementioned works are still rooted in Markov chains. A recent breakthrough is the self-repellent random walk (SRRW) over general graphs (Doshi et al., 2023), a *nonlinear Markov chain* that modifies a time-reversible

Markov chain with $\boldsymbol{\mu}$-invariant transition kernel $\boldsymbol{P}$, as shown by Box ① in Figure 1(a).[1] Unlike non-backtracking random walks relying solely on the most recent history (Alon et al., 2007; Lee et al., 2012), SRRW incorporates the *entire past trajectory* to adaptively adjust transition probabilities towards less-visited states while preserving convergence to $\boldsymbol{\mu}$. Formally, let $\mathcal{G} = (\mathcal{X}, \mathcal{E})$ be a connected, undirected graph with node set $\mathcal{X}$ and edge set $\mathcal{E}$, and let $\mathcal{N}(i) \triangleq \{j \in \mathcal{X} \mid (i, j) \in \mathcal{E}\}$ be the neighbor set of state $i$. Its cardinality is denoted $|\mathcal{N}(i)|$. Let $\overline{\mathcal{N}}(i) \triangleq \mathcal{N}(i) \cup \{i\}$ be the 'expanded' neighborhood of state $i$. Denote by $x_i \in (0, 1)$ the visit frequency of state $i \in \mathcal{X}$. Given a time-reversible kernel $\boldsymbol{P}$ with $\mu_i P_{ij} = \mu_j P_{ji}$ where $P_{ij} > 0$ only if $(i, j) \in \mathcal{E}$, SRRW constructs a *nonlinear* kernel $\boldsymbol{K}[\mathbf{x}]$, parametrized by $\mathbf{x} \triangleq [x_i]_{i \in \mathcal{X}}$, such that, for $j \in \overline{\mathcal{N}}(i)$,

$$K_{ij}[\mathbf{x}] = Z_i^{-1} P_{ij}(x_j/\mu_j)^{-\alpha}, \tag{1}$$

where $\alpha \geq 0$ is the strength of self-repellency, and the normalizing constant $Z_i \triangleq \sum_{k \in \overline{\mathcal{N}}(i)} P_{ik}(x_k/\mu_k)^{-\alpha}$. Note that $K_{ij}[\mathbf{x}] = 0$ for $j \notin \overline{\mathcal{N}}(i)$, conforming to the graph structure. The transition probability to state $j$ is dynamically enlarged or reduced from the original probability $P_{ij}$ based on the history so far via the ratio $(x_j/\mu_j)^{-\alpha}$ between empirical measure and target distribution of state $j$ (Box ② in Figure 1). The kernel (1) is also known to be 'scale-invariant' in the sense that it remains unchanged under any constant multiple of $\mathbf{x}$ or $\boldsymbol{\mu}$. This self-repellent scheme shows remarkable success, achieving near-zero variance for large $\alpha$ at a rate of $O(1/\alpha)$ for graph sampling (Doshi et al., 2023) and $O(1/\alpha^2)$ for distributed optimization (Hu et al., 2024a), and can even outperform *i.i.d.* sampling (i.e., a random jumper) while still walking on the graph.

## 1.2. Limitations of SRRW

**Computational Issue.** Despite its theoretical appeal and general applicability to reversible MCMC samplers, SRRW suffers from critical computational overhead that undermines its practicality, a shortfall overlooked in Doshi et al.

---

[1]In practice, the unnormalized target $\tilde{\boldsymbol{\mu}}$ is typically used in MCMC algorithms instead of the normalized $\boldsymbol{\mu}$, bypassing the need for calculating the normalizing constant.

(2023). To fully understand these challenges, it is important to first clarify the role of self-transition probability $P_{ii}$ in the time-reversible chain $\boldsymbol{P}$ and how it is used in SRRW. In the MH algorithm, from state $i$, another neighbor state $j \in \mathcal{N}(i)$ is proposed with probability $Q_{ij}$ and accepted with probability $A_{ij} \triangleq \min\{1, \mu_j Q_{ji}/\mu_i Q_{ij}\}$. If accepted, the walker moves to $j$, making the transition probability $P_{ij} = Q_{ij} A_{ij}$, keeping it reversible with respect to the given target $\boldsymbol{\mu}$. If rejected, the chain remains at $i$, contributing to the self-transition probability $P_{ii} = 1 - \sum_{j \neq i} Q_{ij} A_{ij}$. Note that $P_{ii} > 0$ unless $A_{ij} = 1$ for all $j \neq i$, a rare case where the proposal $\boldsymbol{Q}$ is already $\boldsymbol{\mu}$-invariant. Importantly, $P_{ii}$ is just a byproduct out of the accept-reject mechanism, whose value is never explicitly pre-computed in standard MH algorithms. In addition, the acceptance ratio $A_{ij}$ is evaluated only for the proposed state $j$, ensuring a lightweight sequential implementation.

However, evaluating $K_{ij}[\mathbf{x}]$ in (1) demands pre-computation of the normalizing constant $Z_i$, which necessitates knowledge of $P_{ij}$ for all $j \in \overline{\mathcal{N}}(i)$, *including* the self-transition probability $P_{ii}$ when $j = i$, implying that all $A_{ij}$ must be pre-calculated. This modification completely *destroys* the essence of MH, where the acceptance ratio is evaluated on demand only for a proposed state one at a time and is never pre-quantified for all possible $j \in \overline{\mathcal{N}}(i)$. This issue becomes even more pronounced in scenarios with a large neighborhood size $|\overline{\mathcal{N}}(i)|$, where the computational cost scales accordingly. On the other hand, sampling $j$ directly from $K_{(i,*)}[\mathbf{x}] \propto P_{(i,*)}(x_*/\mu_*)^{-\alpha}$ as a target distribution over $\overline{\mathcal{N}}(i)$ might be deemed as an alternative to bypass direct computation of the normalizing constant $Z_i$. However, such approach also fails because it still requires knowledge of the target distribution up to a multiplicative constant, including $P_{ii}$, defying the purpose of lightweight sampling. In short, $P_{ii}$ becomes a requirement in SRRW rather than a natural outcome of MH, translating to equivalent computational costs to sample from $K_{(i,*)}[\mathbf{x}]$, and thus offering no benefits.

**Requirement of Time-Reversibility.** Another significant drawback of SRRW is its strict reliance on time-reversible Markov chains. While this reversibility ensures a well-defined stationary distribution for the modified kernel $\boldsymbol{K}[\mathbf{x}]$ (Doshi et al., 2023, Proposition 2.1), it inherently excludes non-reversible MCMC techniques that still converge to the same target distribution $\boldsymbol{\mu}$ with better performance (Neal, 2004; Suwa & Todo, 2010; Lee et al., 2012; Chen & Hwang, 2013; Maire et al., 2014; Ma et al., 2016; Bierkens, 2016; Andrieu & Livingstone, 2021).

**Memory Constraints.** Despite being technically viable, SRRW encounters memory issues in large graphs and configuration spaces where the empirical measure $\mathbf{x}$ shares the dimensionality with the size of the state space. Retaining complete historical data through $\mathbf{x}$ at each step can exceed

memory capacity in simulations. Thus, it is natural to ask:

*Can we design a universal method to harness SRRW benefits, while tackling these challenges: (i) maintain computational efficiency, (ii) leverage both reversible and non-reversible MCMC samplers, and (iii) reduce memory usage?*

### 1.3. Our Approach and Contributions

In this work, we tackle the three aforementioned challenges arising from nonlinear kernels of SRRW. Specifically, we theoretically resolve two of them: computational issue and incompatibility with non-reversible Markov chains, and propose a heuristic scheme that reduces storage requirements for memory issue. Comprehensive applications to exponentially large state spaces, i.e., high-dimensional problems, are deferred to future work.

In SRRW, the self-repellent mechanism is embedded directly into the transition kernel $\boldsymbol{K}[\mathbf{x}]$, leading to significant computational costs, as discussed earlier. However, the essence of a history-dependent scheme need not reside in the kernel itself. As depicted in Box ③ of Figure 1(b), we shift this scheme to a family of *'history-driven' target (HDT) distributions*. Specifically, we replace the original target $\boldsymbol{\mu}$, a single point in the probability simplex, with an adaptive distribution $\boldsymbol{\pi}[\mathbf{x}]$ parameterized by the empirical measure $\mathbf{x}$, which evolves dynamically at each step based on the 'feedback' from past visits. The exact formulation of $\boldsymbol{\pi}[\mathbf{x}]$ in (6) ensures that less/more-visited state $i$ is assigned higher/lower probabilities than the original target $\mu_i$, dynamically over time. Our approach allows for seamless integration with any MCMC technique, using the target $\boldsymbol{\pi}[\mathbf{x}]$ for a given $\mathbf{x}$. Unlike SRRW, in our framework, we can reuse any advanced MCMC technique by simply replacing target $\boldsymbol{\mu}$ by our design $\boldsymbol{\pi}[\mathbf{x}]$ in both reversible and non-reversible samplers, retaining the lightweight, sequential, and adaptive nature of MCMC methods, as shown in Box ④ of Figure 1. By decoupling the self-repellent mechanism from the transition kernel, we overcome the computational limitations of SRRW while achieving near-zero variance. This paradigm shift offers the best of both worlds: computational efficiency and broad compatibility with advanced MCMC samplers. Our contributions are as follows.

1. **A General Self-Repellent Framework:** We introduce the *HDT framework*, which replaces the target $\boldsymbol{\mu}$ by the history-driven target distribution $\boldsymbol{\pi}[\mathbf{x}]$ via rigorous design. This formulation is compatible with both reversible and non-reversible MCMC samplers while eliminating the computational overhead in SRRW.

2. **Theoretical Guarantees:** We prove that our HDT framework converges almost surely to the original target $\boldsymbol{\mu}$ and achieves $O(1/\alpha)$ variance reduction compared to the base sampler in the central limit theorem (CLT). Moreover, we establish a cost-based CLT, show-

ing an additional $O(1/\mathbb{E}[|\overline{\mathcal{N}}(i)|])$ variance reduction relative to SRRW by the average neighborhood size.

3. **Empirical Evaluation:** We perform extensive graph sampling simulations across various settings, including real-world graphs and both reversible and non-reversible MCMC samplers, and showcase the consistent efficiency and adaptability of our HDT framework.

4. **Scalable Implementation for Large Graphs:** We implement an LRU (Least Recently Used) cache scheme to manage $\mathbf{x}$ with limited memory. For example, even when maintaining visit frequencies for only $10\%$ of states, our method leads to more than a $10\%$ reduction in total variation distance over the original MCMC method, highlighting HDT's effectiveness in limited-memory environments.

## 2. Preliminaries

**Basic Notations.** Let $\mathcal{X}$ denote a finite discrete state space. Vectors are denoted by lower-case bold letters, e.g., $\boldsymbol{v} \triangleq [v_i]_{i \in \mathcal{X}}$, and matrices by upper-case bold, e.g., $\boldsymbol{M} \triangleq [M_{ij}]_{i,j \in \mathcal{X}}$. The diagonal matrix $\boldsymbol{D_v}$ is constructed from the vector $\boldsymbol{v}$, with its components placed along the main diagonal. We denote by $\mathbf{1}, \mathbf{0}$ vectors of all ones and zeros with proper dimensions, respectively. Denote by $\Sigma$ the $|\mathcal{X}|$-dimensional probability simplex over $\mathcal{X}$, with $\text{Int}(\Sigma)$ being its interior, i.e., $\mathbf{x} \in \text{Int}(\Sigma)$ implies that $x_i \in (0, 1), \forall i \in \mathcal{X}$. For a probability vector $\mathbf{x} \in \Sigma$, we write $\tilde{\mathbf{x}}$ to denote any of its unnormalized counterparts, i.e., $\mathbf{x} = \tilde{\mathbf{x}}/(\mathbf{1}^T \tilde{\mathbf{x}}) \propto \tilde{\mathbf{x}}$. Define $\boldsymbol{\delta}_i$ as the canonical vector whose components are all zero, except the $i$-th entry being one. Let $N(\mathbf{0}, \boldsymbol{V})$ represent the multivariate Gaussian distribution with zero mean and covariance matrix $\boldsymbol{V}$. We use $\xrightarrow[dist.]{}$ for weak convergence.

**Ergodic Markov Chains.** Consider an ergodic Markov chain with transition kernel $\boldsymbol{P} \in \mathbb{R}^{|\mathcal{X}| \times |\mathcal{X}|}$ and target distribution $\boldsymbol{\mu} \in \text{Int}(\Sigma)$, satisfying $\boldsymbol{P}\mathbf{1} = \mathbf{1}$ and $\boldsymbol{\mu}^T \boldsymbol{P} = \boldsymbol{\mu}^T$. In MCMC, the transition kernel $\boldsymbol{P}$ is defined by the target $\boldsymbol{\mu}$. Throughout this work, we assume that $\boldsymbol{P}$ is full-rank and continuous in $\boldsymbol{\mu}$. Denote by $(\lambda_i, \boldsymbol{u}_i, \boldsymbol{v}_i)$ the eigenpair of $\boldsymbol{P}$, comprising the eigenvalues as well as the corresponding left and right eigenvectors. Here, the Perron–Frobenius eigenvalue $\lambda_1 = 1$ with its corresponding eigenvectors $\boldsymbol{u}_1 = \boldsymbol{\mu}$ and $\boldsymbol{v}_1 = \mathbf{1}$. In addition, $\boldsymbol{u}_i^T \boldsymbol{v}_i = 1$ and $\boldsymbol{u}_i^T \boldsymbol{v}_j = 0$ for all $i, j \in [2, |\mathcal{X}|]$ with $i \neq j$. Consider the sample path $\{X_s\}_{s \geq 1}$ driven by an ergodic Markov chain on $\mathcal{X}$, with $\delta(\cdot)$ as the indicator function. The cumulative visit count to state $i$ by time $n$ is expressed as $\tilde{x}_n(i) \triangleq \sum_{s=1}^{n} \delta(X_s = i)$, ensuring $\sum_{i \in \mathcal{X}} \tilde{x}_n(i) = n$. The empirical measure $\mathbf{x}_n \triangleq [x_n(i)]_{i \in \mathcal{X}}$ is the normalized version of $\tilde{\mathbf{x}}_n$, hence $\mathbf{x}_n = \tilde{\mathbf{x}}_n/(\mathbf{1}^T \tilde{\mathbf{x}}_n) = \tilde{\mathbf{x}}_n/n$. Alternatively, we can express $\mathbf{x}_n$ iteratively as follows:

$$\mathbf{x}_{n+1} = \mathbf{x}_n + \frac{1}{n+1}(\boldsymbol{\delta}_{X_{n+1}} - \mathbf{x}_n). \quad (2)$$

The ergodic theorem for Markov chains (Brémaud, 2013, Theorem 3.3.2) states that the empirical measure $\mathbf{x}_n$ of an ergodic Markov chain, updated via (2), almost surely converges to $\boldsymbol{\mu}$ as $n \to \infty$. In addition, the multivariate CLT (Brooks et al., 2011, Chapter 1.8.1) shows that $\sqrt{n}(\mathbf{x}_n - \boldsymbol{\mu})$ weakly converges to $N(\mathbf{0}, \boldsymbol{V})$, where the covariance matrix

$$\boldsymbol{V} = \lim_{t \to \infty} \tfrac{1}{t}\mathbb{E}[(\sum_{s=1}^{t}(\boldsymbol{\delta}_{X_s} - \boldsymbol{\mu}))(\sum_{s=1}^{t}(\boldsymbol{\delta}_{X_s} - \boldsymbol{\mu}))^T], \quad (3)$$

which serves as the covariance matrix of both reversible and non-reversible MCMC samplers in Theorem 3.3. By Brémaud (2013, Chapter 6.3.3), $\boldsymbol{V}$ can be rewritten as

$$\boldsymbol{V} = \sum_{i=2}^{|\mathcal{X}|} \frac{1 + \lambda_i}{1 - \lambda_i} \boldsymbol{D_\mu} \boldsymbol{v}_i \boldsymbol{u}_i^T. \quad (4)$$

When $\boldsymbol{P}$ is reversible, the property $\boldsymbol{u}_i = \boldsymbol{D_\mu} \boldsymbol{v}_i$ gives $\boldsymbol{V} = \sum_{i=2}^{|\mathcal{X}|} \frac{1+\lambda_i}{1-\lambda_i} \boldsymbol{u}_i \boldsymbol{u}_i^T$. See Appendix A for the derivation of (4).

**Properties of SRRW.** The SRRW algorithm in Doshi et al. (2023) includes two steps at each time $n$: First, sample $X_{n+1} \sim K_{(X_n, \cdot)}[\mathbf{x}_n]$ in (1) using the current empirical measure $\mathbf{x}_n$ and state $X_n$; Second, update $\mathbf{x}_{n+1}$ via (2). This ensures that the SRRW kernel $\boldsymbol{K}[\mathbf{x}]$ is adapted to the updated empirical measure $\mathbf{x}_n$ at each time $n$, making it a nonlinear Markov chain. A notable property of SRRW is 'scale-invariance', i.e., for any non-zero scalar $C$, $K_{ij}[C\mathbf{x}] = K_{ij}[\mathbf{x}], \forall i, j \in \mathcal{X}$, such that computing $K_{ij}[\mathbf{x}]$ only requires knowing vectors $\mathbf{x}, \boldsymbol{\mu}$ up to some constant multiples. It has been theoretically proved that the empirical measure $\mathbf{x}_n \to \boldsymbol{\mu}$ almost surely as $n \to \infty$, and the scaled error $\sqrt{n}(\mathbf{x}_n - \boldsymbol{\mu}) \xrightarrow[dist.]{n \to \infty} N(\mathbf{0}, \boldsymbol{V}^{\text{SRRW}}(\alpha))$, where

$$\boldsymbol{V}^{\text{SRRW}}(\alpha) = \sum_{i=2}^{|\mathcal{X}|} \frac{1}{2\alpha(\lambda_i + 1) + 1} \cdot \frac{1 + \lambda_i}{1 - \lambda_i} \boldsymbol{u}_i \boldsymbol{u}_i^T, \quad (5)$$

and $\lambda_i, \boldsymbol{u}_i$ are eigenvalues and left eigenvectors of the time-reversible base MCMC kernel $\boldsymbol{P}$ leveraged by SRRW.

## 3. Main Results

We present our theoretical findings by first outlining the design principles for a HDT distribution that balances exploration with computational efficiency. Next, we examine the convergence and statistical properties of the HDT-MCMC algorithm. Lastly, we evaluate its computational cost against SRRW within a fixed budget using a cost-based CLT, showcasing the performance benefits of HDT-MCMC.

### 3.1. Design of History-Driven Target Distribution

The foundation of HDT-MCMC is in the design of a history-driven target $\boldsymbol{\pi}[\mathbf{x}, \boldsymbol{\mu}]$ that relies solely on the visit count $\tilde{\mathbf{x}}$ and an unnormalized target $\tilde{\boldsymbol{\mu}}$ (as an input), while encapsulating self-repellent behavior in the resulting kernel. Our goal is to design the HDT $\boldsymbol{\pi}[\mathbf{x}, \boldsymbol{\mu}]$ to satisfy the following four conditions:

C1 (Scale Invariance). $\pi_i[\mathbf{x}, \boldsymbol{\mu}] = \pi_i[\tilde{\mathbf{x}}, \tilde{\boldsymbol{\mu}}]$.

C2 (Local Dependence). The unnormalized term $\tilde{\pi}_i[\mathbf{x}, \boldsymbol{\mu}]$ depends only on $\mu_i$ and $x_i$, and is continuous in $x_i, \mu_i$.

Henceforth, we suppress $\boldsymbol{\mu}$ and simply write $\boldsymbol{\pi}[\mathbf{x}]$ for brevity, whenever its dependence on $\boldsymbol{\mu}$ is clear from the context.

C3 (Fixed Point). $\boldsymbol{\pi}[\boldsymbol{\mu}] = \boldsymbol{\mu}$.

C4 (History Dependence). $\boldsymbol{\pi}[\mathbf{x}]$ promotes/deters exploration of under/over-sampled states.

C1 preserves the key property that our HDT-MCMC algorithm only needs the unnormalized terms $\tilde{\boldsymbol{\mu}}$, and $\tilde{\mathbf{x}}$ when, for instance, computing an acceptance ratio of MH. Examples of the unnormalized target $\tilde{\boldsymbol{\mu}}$ include: $\tilde{\mu}_i = 1$ for uniform target, $\tilde{\mu}_i = |\mathcal{N}(i)|$ for degree-based target, and $\tilde{\mu}_i = e^{-H(i)}$ for energy-specific target (Gjoka et al., 2011; Lee et al., 2012; Grathwohl et al., 2021; Pynadath et al., 2024). C2 ensures that the sampler uses only minimal local information, i.e., $\tilde{x}_i$ and $\tilde{\mu}_i$, in its update. By eliminating neighbor information, the algorithm avoids prohibitive computational costs associated with increasing neighborhood size, e.g., in large or complete graphs. C3 indicates that if empirical measure $\mathbf{x}$ converges to the target $\boldsymbol{\mu}$, the HDT $\boldsymbol{\pi}[\boldsymbol{\mu}]$ remains $\boldsymbol{\mu}$ itself. Indeed, if $\boldsymbol{\pi}[\boldsymbol{\mu}] \neq \boldsymbol{\mu}$, then an MCMC sampler with its target $\boldsymbol{\pi}[\mathbf{x}]$ would converge to a different distribution, contradicting our goal of preserving $\boldsymbol{\mu}$ as the true target distribution. C4 is to facilitate exploration, ensuring that the sampler adaptively 'pushes' itself away from states already visited more often than $\mu_i$, thereby mimicking the key self-repellent effect of SRRW in Doshi et al. (2023).

**Lemma 3.1.** *Conditions C1 - C4 hold if and only if*

$$\pi_i[\mathbf{x}] \propto \mu_i (x_i/\mu_i)^{-\alpha} \quad \text{for any } \alpha > 0. \tag{6}$$

The proof of Lemma 3.1 can be found in Appendix B. It reveals that an HDT distribution $\boldsymbol{\pi}[\mathbf{x}]$ satisfying all four conditions C1 - C4 must take the simple form of (6). A special case $\alpha = 0$ reduces $\boldsymbol{\pi}[\mathbf{x}]$ to the original target $\boldsymbol{\mu}$, which becomes history-independent. Lemma 3.1 also demonstrates that our design (6) suffices to work with unnormalized quantities $\tilde{\boldsymbol{\mu}}$ and $\tilde{\mathbf{x}}$ in place of $\boldsymbol{\mu}$ and $\mathbf{x}$, i.e.,

$$\pi_i[\mathbf{x}] \propto \tilde{\pi}_i[\mathbf{x}] = \tilde{\mu}_i (\tilde{x}_i/\tilde{\mu}_i)^{-\alpha}. \tag{7}$$

Following Lemma 3.1, Algorithm 1 shows the steps in our HDT-MCMC framework.[2] As illustrated by Box ④ in Figure 1, the base MCMC sampler for graph sampling in our framework can either be time-reversible, i.e., MH (Metropolis et al., 1953; Hastings, 1970), MHDR (Green & Mira, 2001), MTM (Liu et al., 2000; Chang et al., 2022), or non-reversible, i.e., MHDA (Lee et al., 2012), 2-cycle Markov chains (Maire et al., 2014; Andrieu & Livingstone, 2021), and non-reversible MH (Bierkens, 2016; Thin et al., 2020).

---

[2]Similar to Doshi et al. (2023), we initialize $\tilde{x}_i > 0$ with fake visit count for well-defined $\boldsymbol{\pi}[\mathbf{x}]$ at every step. In reality, memory is allocated for states that have received actual visits, enabling adaptive storage for $\tilde{\mathbf{x}}$.

---

**Algorithm 1** HDT-MCMC: Graph Sampling Framework

---

**Input:** Graph $\mathcal{G}(\mathcal{X}, \mathcal{E})$, parameter $\alpha \geq 0$, unnormalized target $\tilde{\boldsymbol{\mu}}$, number of iterations $T$, a base MCMC sampler (*Bring Your Own MCMC*).
**Initialization:** state $X_0 \in \mathcal{X}$, visit count $\tilde{x}(i) > 0, \forall i \in \mathcal{X}$.
**for** $n = 0$ **to** $T - 1$ **do**
    Step 1: Use the base sampler (reversible or non-reversible) to draw $X_{n+1}$ with history-driven target $\boldsymbol{\pi}[\mathbf{x}]$ from (7).
    Step 2: Update visit count $\tilde{x}(X_{n+1}) \leftarrow \tilde{x}(X_{n+1}) + 1$;
**end for**
**Output:** A set of samples $\{X_n\}_{n=1}^T$.

---

As an example, we here illustrate our HDT-MCMC if we use the standard MH algorithm as the base MCMC sampler (Step 1) in Algorithm 1.[3] At current state $X_n = i$, a candidate $j \in \mathcal{N}(i)$ is selected with probability $Q_{ij}$, then the acceptance ratio is calculated through

$$A_{ij}[\mathbf{x}] = \min\left\{1, \frac{\pi_j[\mathbf{x}]Q_{ji}}{\pi_i[\mathbf{x}]Q_{ij}}\right\} = \min\left\{1, \frac{\tilde{\mu}_j(\tilde{x}_j/\tilde{\mu}_j)^{-\alpha}Q_{ji}}{\tilde{\mu}_i(\tilde{x}_i/\tilde{\mu}_i)^{-\alpha}Q_{ij}}\right\}, \tag{8}$$

where only the unnormalized terms $\tilde{\mu}_i, \tilde{\mu}_j, \tilde{x}_i$, and $\tilde{x}_j$ are required, keeping the same computational cost as standard MH with true target $\boldsymbol{\mu}$. Then, the sampler accepts state $j$ with probability $A_{ij}[\mathbf{x}]$ and sets $X_{n+1} = j$, or rejects it with probability $1 - A_{ij}[\mathbf{x}]$ upon which $X_{n+1} = i$ and repeats the procedure. Note that we recover the acceptance ratio $A_{ij}$ of the standard MH with target $\boldsymbol{\mu}$, when $\alpha = 0$. This manner alters the target $\boldsymbol{\mu}$ of the standard MH algorithm by HDT $\boldsymbol{\pi}[\mathbf{x}]$ while preserving the lightweight, on-demand nature of MH, in contrast to SRRW in which $P_{ij}$ must be evaluated for all possible $j \in \overline{\mathcal{N}}(i)$ for a sample $X_{n+1}$. In addition to computational efficiency, $A_{ij}[\mathbf{x}]$ inherently embeds the 'self-repellent' effect. If state $j$ is relatively less-visited than state $i$, i.e., $\tilde{x}_j/\tilde{\mu}_j < \tilde{x}_i/\tilde{\mu}_i$, it then follows that $A_{ij}[\mathbf{x}] \geq A_{ij}$, implying that state $j$ is more likely to be accepted than the case with standard MH, and vice versa.

*Remark* 3.1. In Doshi et al. (2023), SRRW directly modifies a time-reversible Markov chain $\boldsymbol{P}$ to incorporate self-repellency into the kernel $\boldsymbol{K}[\mathbf{x}]$ as in (1), which is then shown to be, for any given $\mathbf{x} \in \text{Int}(\Sigma)$, reversible w.r.t.

$$\pi_i^{\text{SRRW}}[\mathbf{x}] \propto \mu_i(x_i/\mu_i)^{-\alpha} \sum_{j \in \overline{\mathcal{N}}(i)} P_{ij}(x_j/\mu_j)^{-\alpha}, \forall i \in \mathcal{X},$$

whose proof in Doshi et al. (2023, Appendix A) critically depends on the reversibility of $\boldsymbol{P}$ w.r.t. $\boldsymbol{\mu}$. Note that $\pi_i^{\text{SRRW}}[\mathbf{x}]$ is the byproduct of the constructed kernel $\boldsymbol{K}[\mathbf{x}]$ as in (1), thus inheriting the same neighborhood dependency. Simply adopting $\boldsymbol{\pi}^{\text{SRRW}}[\mathbf{x}]$ in our Algorithm 1 violates C2 and would incur high computational cost for resulting nonlinear

---

[3]Detailed overview of Algorithm 1 applied to advanced MCMC samplers are in Appendix C.

kernels. In contrast, our HDT (6) *decouples* neighbors in the target distribution itself, which eliminates the need to evaluate transition probabilities for all neighbors at each step, offering substantial computational savings and compatibility with both reversible and non-reversible samplers.

## 3.2. Performance of HDT-MCMC

We next analyze HDT-MCMC regarding (i) almost-sure convergence of the empirical measure $\mathbf{x}_n$ to $\boldsymbol{\mu}$, and (ii) an $O(1/\alpha)$ variance reduction relative to the base MCMC sampler with true target $\boldsymbol{\mu}$.

Observe that (2) allows us to decompose $\mathbf{x}_{n+1}$ as

$$\mathbf{x}_{n+1} = \mathbf{x}_n + \frac{1}{n+1}[\underbrace{(\boldsymbol{\pi}[\mathbf{x}_n] - \mathbf{x}_n)}_{\text{deterministic drift}} + \underbrace{(\boldsymbol{\delta}_{X_{n+1}} - \boldsymbol{\pi}[\mathbf{x}_n])}_{\text{noise term}}],$$

which is the standard step in stochastic approximation (SA) with controlled Markovian dynamics (Kushner & Yin, 2003; Benveniste et al., 2012; Borkar, 2022). It can be viewed as combining a deterministic drift $\boldsymbol{\pi}[\mathbf{x}_n] - \mathbf{x}_n$ towards the solution of the ODE

$$\dot{\mathbf{x}}(t) = \boldsymbol{\pi}[\mathbf{x}(t)] - \mathbf{x}(t) \qquad (9)$$

and a noise term $\boldsymbol{\delta}_{X_{n+1}} - \boldsymbol{\pi}[\mathbf{x}_n]$. The ODE viewpoint clarifies the global asymptotic stability of $\boldsymbol{\mu}$, while the noise term characterizes fluctuations around $\boldsymbol{\mu}$.

**Lemma 3.2.** *For $\boldsymbol{\pi}[\mathbf{x}]$ in (6), the ODE (9) has a unique fixed point $\boldsymbol{\mu}$, which is globally asymptotically stable.*

Proofs follow by showing that target $\boldsymbol{\mu}$ is the unique stable equilibrium and employing standard Lyapunov stability theory, as detailed in Appendix D. For the iteration $\mathbf{x}_n$ in (2), we need to additionally account for the noise term driven by a history-dependent MCMC from Algorithm 1. We impose the following assumption on $\mathbf{x}_n$ throughout the paper.

**Assumption 1.** $\mathbf{x}_n \in \text{Int}(\Sigma)$ for all $n \geq 0$ almost surely.

Assumption 1 guarantees $x_n(i) > 0$ almost surely for all $i \in \mathcal{X}$ and $n \geq 0$, keeping $\boldsymbol{\pi}[\mathbf{x}]$ well-defined at each step. This type of assumption is standard in the SA literature (Fort, 2015; Borkar, 2022; Li et al., 2023). In practice, it can be enforced using truncation-based methods within (2), as discussed in Doshi et al. (2023, Remark 4.5 and Appendix E). These methods effectively prevent any component $x_n(i)$ from approaching zero, thus maintaining $\mathbf{x}_n \in \text{Int}(\Sigma)$. Under this assumption, we obtain the following:

**Theorem 3.3** (Ergodicity and CLT). *HDT-MCMC in Algorithm 1 satisfies*

*(a) $\mathbf{x}_n \to \boldsymbol{\mu}$ almost surely as $n \to \infty$.*

*(b) $\sqrt{n}(\mathbf{x}_n - \boldsymbol{\mu}) \xrightarrow[\text{dist.}]{n \to \infty} N(\mathbf{0}, \boldsymbol{V}^{HDT}(\alpha))$, where*
$$\boldsymbol{V}^{HDT}(\alpha) = \frac{1}{2\alpha + 1}\boldsymbol{V}^{base}, \qquad (10)$$
*and $\boldsymbol{V}^{base}$ is the limiting covariance of the base MCMC sampler (reversible or non-reversible) with target $\boldsymbol{\mu}$ in (3).*

The full proof of Theorem 3.3 is in Appendix E, where we leverage the existing asymptotic analysis from the SA literature (Delyon et al., 1999; Fort, 2015), similarly used in Doshi et al. (2023, Appendix C). However, the primary technical challenge arises from handling non-reversible Markov chains, which is excluded from Doshi et al. (2023) by the nature of their kernel design. We proceed with our analysis that is specifically tailored to the augmented state space on which the non-reversible Markov chain is defined, and solve a mismatch issue between the augmented space and the original space by only tracking the marginal empirical measure $\mathbf{x} \in \text{Int}(\Sigma)$ in the original space $\mathcal{X}$. Theorem 3.3 highlights two appealing features of HDT-MCMC: (i) It preserves the unbiased sampling by converging to the true target $\boldsymbol{\mu}$. (ii) It offers an $O(1/\alpha)$ variance reduction, achieving a similar near-zero variance phenomenon of SRRW but without additional computational overhead or the need for time-reversibility. When $\alpha = 0$, we recover the baseline scenario $\boldsymbol{\pi}[\mathbf{x}] \equiv \boldsymbol{\mu}$ and $\boldsymbol{V}^{HDT}(0) = \boldsymbol{V}^{base}$, as expected.

Moreover, Theorem 3.3 leads to an instant result as follows.

**Corollary 3.4.** *Suppose two MCMC samplers $S_1$ and $S_2$ converge to $\boldsymbol{\mu}$ with limiting covariances $\boldsymbol{V}^{S_1}$ and $\boldsymbol{V}^{S_2}$ satisfying $\boldsymbol{V}^{S_1} \preceq \boldsymbol{V}^{S_2}$.[4] Applying HDT framework to both, yielding $\boldsymbol{V}^{S_1\text{-}HDT}(\alpha)$ and $\boldsymbol{V}^{S_2\text{-}HDT}(\alpha)$, preserves the ordering:*

$$\boldsymbol{V}^{S_1\text{-}HDT}(\alpha) \preceq \boldsymbol{V}^{S_2\text{-}HDT}(\alpha), \quad \forall \alpha \geq 0.$$

The proof is straightforward from (10). Hence, any known covariance orderings between reversible and non-reversible samplers (see Lee et al. (2012); Maire et al. (2014); Bierkens (2016); Andrieu & Livingstone (2021)) carry over to our HDT-MCMC framework, whereas SRRW cannot accommodate non-reversible Markov chains.

## 3.3. Comparative Analysis of Computational Costs

We now compare the performance of our HDT-MCMC to SRRW (Doshi et al., 2023) under a fixed total computational budget $B$. Although both methods achieve an $O(1/\alpha)$ variance reduction, HDT-MCMC requires significantly less computation per sample.

Because SRRW mandates a time-reversible base MCMC sampler, we restrict our comparison to the same reversible chain (as illustrated in Boxes ② and ④ of Figure 1). Let $a_i$ (*resp.* $b_i$) $\in (0, \infty)$ be the computational cost of the $i$-th sample in HDT-MCMC (*resp.* SRRW). Define:
$$T^{HDT}(B) \triangleq \max\{k \mid a_1 + a_2 + \cdots + a_k \leq B\}$$
$$T^{SRRW}(B) \triangleq \max\{k' \mid b_1 + b_2 + \cdots + b_{k'} \leq B\}$$

so that $T^{HDT}(B)$ (*resp.* $T^{SRRW}(B)$) represents the total number of samples that HDT-MCMC (*resp.* SRRW) can generate before hitting the budget $B$. Intuitively, under the same

---

[4] Two symmetric matrices $M_1$, $M_2$ follow $M_1 \preceq M_2$ (or $M_1 \succeq M_2$) if $M_2 - M_1$ is positive (or negative) semi-definite.

budget, SRRW's higher per-sample cost yields fewer total samples. To quantify this effect, we now compare these two frameworks under the same but large amount of total budget $B$ instead of the number of samples as done in Section 3.2.

Clearly, we have $T^{\text{HDT}}(B) \to \infty$ and $T^{\text{SRRW}}(B) \to \infty$ as $B \to \infty$. Let $\mathbf{x}_n$ (resp. $\mathbf{y}_n$) be the empirical measure of HDT-MCMC (resp. SRRW). As $B \to \infty$, both $\mathbf{x}_{T^{\text{HDT}}(B)}$ and $\mathbf{y}_{T^{\text{SRRW}}(B)}$ still converge almost surely to $\boldsymbol{\mu}$, an instant result of our Theorem 3.3(a) and Doshi et al. (2023, Theorem 4.1). Analogous to our Theorem 3.3, rather than evaluating at $n \to \infty$, we quantify how HDT-MCMC and SRRW behave under $B \to \infty$ via the following cost-based CLT:

**Theorem 3.5** (Cost-Based CLT). *Suppose that as $B \to \infty$,*

$$B/T^{\text{HDT}}(B) \to C^{\text{HDT}}, \quad B/T^{\text{SRRW}}(B) \to C^{\text{SRRW}} \quad \text{a.s.}$$

*Then, we have*

$$\sqrt{B}(\mathbf{x}_{T^{\text{HDT}}(B)} - \boldsymbol{\mu}) \xrightarrow[\text{dist.}]{B \to \infty} N(\mathbf{0}, C^{\text{HDT}} \boldsymbol{V}^{\text{HDT}}(\alpha)) \quad (11)$$

$$\sqrt{B}(\mathbf{y}_{T^{\text{SRRW}}(B)} - \boldsymbol{\mu}) \xrightarrow[\text{dist.}]{B \to \infty} N(\mathbf{0}, C^{\text{SRRW}} \boldsymbol{V}^{\text{SRRW}}(\alpha)) \quad (12)$$

*where $\boldsymbol{V}^{\text{HDT}}(\alpha)$ is given by (10), and $\boldsymbol{V}^{\text{SRRW}}(\alpha)$ by (5).*

We leverage the random-change-of-time theory (Billingsley, 2013) and Slutsky's theorem (Ash & Doléans-Dade, 2000) to transform our time-based CLT (Theorem 3.3) to the cost-based CLT (Theorem 3.5), with details in Appendix F.

In practice, HDT-MCMC's per-sample cost $C^{\text{HDT}}$ can be significantly smaller than $C^{\text{SRRW}}$ because the latter must pre-compute the transition probability $K_{ij}[\mathbf{x}]$ at each step. As a concrete example, we focus on the MH framework as the base sampler utilized in many advanced MCMC schemes (Liu et al., 2000; Green & Mira, 2001; Lee et al., 2012; Bierkens, 2016; Zanella, 2020; Chang et al., 2022). Assuming that computing the proposal probability $Q_{ij}$ and inquiring $\tilde{\mu}_i$ incurs $c$ units of cost per each pair $(i, j) \in \mathcal{E}$, it requires $2c$ costs for $A_{ij}[\mathbf{x}]$ in (8). Thus, HDT-MCMC spends $2c$ per sample, whereas SRRW incurs $2c|\overline{\mathcal{N}}(i)|$ subject to state $i$, as discussed in Section 1.2. The following lemma shows the ordering of their cost-based covariances:

**Lemma 3.6.** *The cost-based covariances between SRRW and HDT-MCMC in (11) and (12) are ordered as follows:*

$$C^{\text{HDT}} \boldsymbol{V}^{\text{HDT}}(\alpha) \preceq (2/\mathbb{E}_{i \sim \boldsymbol{\mu}}[|\overline{\mathcal{N}}(i)|]) \cdot C^{\text{SRRW}} \boldsymbol{V}^{\text{SRRW}}(\alpha).$$

See Appendix G for the proof. Lemma 3.6 implies that the cost-based covariance of HDT-MCMC is at least a factor of $2/\mathbb{E}[|\overline{\mathcal{N}}(i)|]$ times smaller than that of SRRW in Loewner ordering for each $\alpha$, suggesting a universal advantage. This factor becomes more pronounced in dense or nearly complete graphs, where the average neighborhood size is $\mathbb{E}[|\overline{\mathcal{N}}(i)|] \gg 2$.

For unbiased graph sampling, at each time $T$, the sampling agent records state $X_T$, obtains the value $f(X_T) \in \mathbb{R}$, and

updates the unbiased MCMC estimator $\psi_T(f)$, aiming to approximate the ground truth $\bar{f}$, such as a global attribute of the unknown graph, and their expressions are defined in the following:

$$\psi_T(f) \triangleq \frac{1}{T} \sum_{s=1}^{T} f(X_s), \quad \bar{f} \triangleq \sum_{i \in \mathcal{X}} \mu_i f(i).$$

Equivalently, we can rewrite the MCMC estimator as $\psi_T(f) = \boldsymbol{f}^T \mathbf{x}_T$ since $\mathbf{x}_T = \frac{1}{T} \sum_{s=1}^{T} \boldsymbol{\delta}_{X_s}$ and $\boldsymbol{f}^T \boldsymbol{\delta}_i = f(i)$, where $\boldsymbol{f} \triangleq [f(i)]_{i \in \mathcal{X}} \in \mathbb{R}^{|\mathcal{X}|}$. Thus, the cost-based variance of $\psi_T(f)$ for HDT-MCMC and SRRW is derived by left multiplying $\boldsymbol{f}^T$ into (11) and (12), yielding

$$\sqrt{T}(\psi_T^{\text{HDT}}(f) - \bar{f}) \xrightarrow[\text{dist.}]{T \to \infty} N(\mathbf{0}, \text{Var}^{\text{HDT}}(\alpha)), \quad (13)$$

$$\sqrt{T}(\psi_T^{\text{SRRW}}(f) - \bar{f}) \xrightarrow[\text{dist.}]{T \to \infty} N(\mathbf{0}, \text{Var}^{\text{SRRW}}(\alpha)), \quad (14)$$

where $\text{Var}^{\text{HDT}}(\alpha) = C^{\text{HDT}} \boldsymbol{f}^T \boldsymbol{V}^{\text{HDT}}(\alpha) \boldsymbol{f}$, and $\text{Var}^{\text{SRRW}}(\alpha) = C^{\text{SRRW}} \boldsymbol{f}^T \boldsymbol{V}^{\text{SRRW}}(\alpha) \boldsymbol{f}$, respectively. By Lemma 3.6 and the Loewner ordering in footnote 4, we have for any $\alpha > 0$,

$$\text{Var}^{\text{HDT}}(\alpha) \leq (2/\mathbb{E}[|\overline{\mathcal{N}}(i)|]) \cdot \text{Var}^{\text{SRRW}}(\alpha),$$

which translates into at least a factor of $2/\mathbb{E}[|\overline{\mathcal{N}}(i)|]$ times smaller cost-based variance of the MCMC estimator $\psi_T(f)$ than the case with SRRW. In addition, unlike SRRW, HDT-MCMC also accommodates non-reversible base samplers (see Corollary 3.4), further enhancing efficiency in many applications.

## 4. Simulations

We design a series of experiments to evaluate the performance of HDT-based MCMC methods in graph sampling tasks. Our goal is to compare HDT-MCMC with various advanced MCMC algorithms, including both reversible and non-reversible Markov chains, and compare HDT-MCMC with SRRW within the same total computational budget.

### 4.1. Simulation Setup

We conduct experiments on two real-world graphs, i.e., facebook (4039 nodes with 88234 edges) and p2p-Gnutella04 (10876 nodes with 39994 edges) from SNAP (Leskovec & Krevl, 2014). We use a uniform target distribution $\boldsymbol{\mu} = \frac{1}{|\mathcal{X}|} \mathbf{1}$ throughout this section while deferring the experiments of non-uniform target to Appendix H.6. In the reversible setting, we apply the standard MH algorithm (MHRW) and MTM with locally balanced weights and $K = 3$ proposed candidates (Chang et al., 2022). In the non-reversible setting, we adopt MHDA (Lee et al., 2012). Additional experiments on WikiVote, p2p-Gnutella08, and non-reversible 2-cycle Markov chains appear in Appendix H. To assess conver-

*Table 1.* Mean (std error) TVD and NRMSE at $n = 15,000$ steps on facebook graph under different $X_0$ groups and fake visit counts $\mathbf{x}_0$. All std error values are in units of $10^{-3}$.

| | Initial State ($X_0$) | | | | Fake Count ($\mathbf{x}_0$) | | | | | |
| | Low Deg | | High Deg | | Deg | | Non-unif | | Unif | |
| Method | TVD | NRMSE | TVD | NRMSE | TVD | NRMSE | TVD | NRMSE | TVD | NRMSE |
|---|---|---|---|---|---|---|---|---|---|---|
| HDT-MHRW | 0.372 (1.221) | 0.027 | 0.371 (1.296) | 0.029 | 0.371 (1.246) | 0.028 | 0.371 (1.281) | 0.028 | 0.371 (1.246) | 0.028 |
| HDT-MTM | 0.283 (1.199) | 0.070 | 0.284 (1.280) | 0.068 | 0.288 (1.751) | 0.098 | 0.285 (1.456) | 0.047 | 0.285 (1.496) | 0.062 |
| HDT-MHDA | 0.364 (1.230) | 0.026 | 0.365 (1.292) | 0.028 | 0.365 (1.281) | 0.027 | 0.365 (1.246) | 0.028 | 0.366 (1.258) | 0.027 |

gence, we use the total variation distance

$$\text{TVD}(\mathbf{x}_n, \boldsymbol{\mu}) \triangleq \frac{1}{2}\|\mathbf{x}_n - \boldsymbol{\mu}\|_1$$

for the distance between the empirical measure $\mathbf{x}_n$ of the collected samples and the target $\boldsymbol{\mu}$. We also evaluate normalized root mean squared error

$$\text{NRMSE}(\psi_n, \bar{\psi}) = \sqrt{\mathbb{E}[(\psi_n(f) - \bar{\psi})^2]} / \bar{\psi}$$

for graph-based group-size estimation with test function $f$ defined in Appendix H.2. Each experiment consists of 1000 independent runs, and one-third of the total iterations is used as the burn-in period.

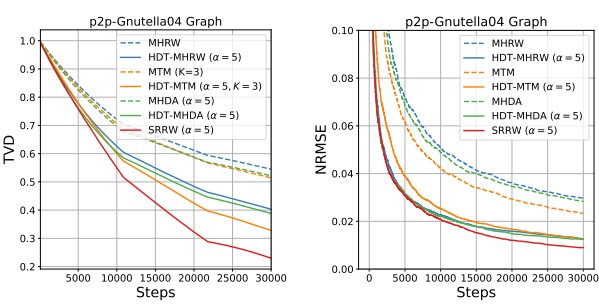

*Figure 2.* TVD and NRMSE Comparison of HDT-MCMC with base chain MHRW, MTM, and MHDA.

### 4.2. Comparison of Base MCMC and its HDT Version

We first compare each base MCMC algorithm with its HDT-enhanced version, setting $\alpha = 5$ in the target $\boldsymbol{\pi}[\mathbf{x}]$ in (6). Figure 2 shows the average TVD and NRMSE for MTM and MHDA, with MHRW serving as a benchmark. In both cases, MTM and MHDA outperform MHRW when targeting $\boldsymbol{\mu}$, which aligns with the theoretical results in Chang et al. (2022) and Lee et al. (2012). Moreover, the HDT-enhanced versions (HDT-MTM and HDT-MHDA) consistently attain lower TVD and NRMSE than their respective base algorithms, indicating faster convergence to $\boldsymbol{\mu}$, consistent with Theorem 3.3. A similar trend is observed in other graphs using TVD and NRMSE metrics (see Appendix H.2). Notably, SRRW (with MHRW as its base) achieves the lowest TVD and NRMSE among all methods but entails substantially higher computational overhead to obtain one sample, whose

effect is not reflected in the number of steps. We shall examine SRRW's performance under fixed budget in the next experiment. We do not combine SRRW with MTM in this experiment due to its heavy computation in $P_{ij}$ for all possible combinations of $K$ intermediate proposed candidates, whereas HDT integrates seamlessly into MTM without additional cost. Moreover, we conduct the experiment on the effect of different $\alpha$ values influence HDT-MHRW algorithm in Appendix H.4 and observe consistent improvement with smaller TVD using larger $\alpha$.

### 4.3. Robustness to Different Initializations

To demonstrate that HDT-MCMC is robust to different initializations, we evaluate its performance when initialized at various nodes, using base chains MHRW, MTM, and MHDA for the HDT-enhanced versions. We show the experimental results for the Facebook graph in Table 1 and include the results for other graphs in Appendix H.3. In particular, we examine the effects of both initial state $X_0$ and the fake visit counts $\mathbf{x}_0$. The initiate state $X_0$ is randomly chosen from either low-degree group (where the node's degree is smaller than the average degree) or high-degree group. This is to test the sensitivity of our algorithm starting from sparse or dense regions. On the other hand, fake visit count $\mathbf{x}_0$ is initialized using one of the following settings: 'Deg' (proportional to node degree, e.g., $\tilde{x}_i = |\mathcal{N}(i)|$ for all $i \in [N]$), 'Non-unif' (a non-uniform draw from a Dirichlet distribution with default hyperparameter 0.5), and 'Unif' (same initial count across all nodes, e.g., $\tilde{x}_i = 1$ for all $i \in [N]$). In Table 1, both TVD and NRMSE[5] show consistent performance when starting from either the low-degree group or the high-degree group. Similarly, a robust result is observed when using different settings of fake visit counts.

### 4.4. Computational Cost Comparison with SRRW

We compare HDT-MHRW and SRRW under a fixed computational budget $B$. At each iteration, SRRW needs to compute or retrieve transition probabilities for every neighbor due to the nature of the kernel design (1), whereas HDT only updates the proposal for a single candidate. This com-

---

[5]NRMSE inherently measures the deviation from the true value, hence confidence intervals are not provided for this metric.

putational cost aligns with the concerns of the $O(\mathcal{N}(i))$ evaluations in the proposal distribution highlighted in Zanella (2020); Grathwohl et al. (2021) in high-dimensional spaces. In graph sampling, simply counting samples to assess performance can be misleading under rate-limited API constraints, e.g., online social network sampling (Xu et al., 2017; Li et al., 2019). The performance gap between HDT-MHRW and SRRW in budget $B$ increases with the degree of the node, as proved in Lemma 3.6. Figure 3 shows that under the same total budget, HDT-MHRW consistently achieves lower TVD and NRMSE than SRRW. The discrepancy is even more pronounced in the Facebook graph in both plots, where the average degree (43.6) far exceeds that of p2p-Gnutella04 (7.4), supporting our discussion after Lemma 3.6, where larger neighborhood size leads to more performance advantage of HDT-MCMC compared to SRRW.

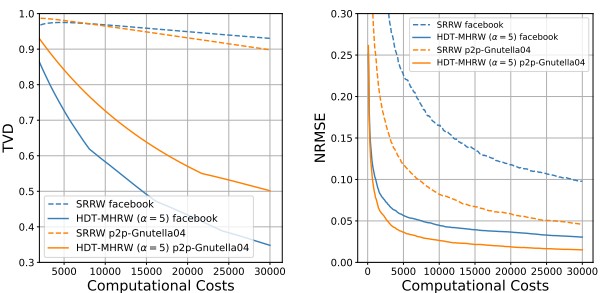

*Figure 3.* TVD and NRMSE Comparison between HDT-MHRW and SRRW under budget constraints.

### 4.5. Least Recently Used (LRU) Cache Scheme

A core challenge in HDT-MCMC, as noted in Section 1.2, involves handling large graphs or exponentially growing configuration state spaces, where storing the entire visit counts $\tilde{\mathbf{x}} \in \mathbb{R}^{|\mathcal{X}|}$ of $|\mathcal{X}|$ dimension becomes infeasible. To address this challenge, we propose a memory-efficient Least Recently Used (LRU) cache strategy and emulate its effect on real-world graphs that serves as a pilot study prior to full deployment in large configuration spaces (beyond this paper's scope).

The essential idea is to track only recently visited states, discarding the least-recently used when capacity in cache $\mathcal{C}$ is reached. This leverages temporal locality, as non-neighboring states do not affect self-repellency. Unlike sparsification methods (Verma et al., 2024; Barnes et al., 2024), which ignore the walker's position, LRU's local simplicity aligns with our HDT-MCMC. For a neighbor $j \notin \mathcal{C}$ of current state $i$, we approximate its frequency via

$$\hat{x}_j = \tilde{\mu}_j |\overline{\mathcal{N}}(i) \cap \mathcal{C}|^{-1} \sum_{k \in \overline{\mathcal{N}}(i) \cap \mathcal{C}} \tilde{x}_k / \tilde{\mu}_k. \quad (15)$$

This approach approximates $\tilde{x}_j / \tilde{\mu}_j$, which mimics (6) via temporal closeness among states frequently visited around $i$, allowing the sampler to maintain self-repellency without

true visit count. We examine HDT-MCMC using an LRU cache with size $|\mathcal{C}| = r|\mathcal{X}|$ where $0 < r < 1$. In Figure 4, the average TVD of HDT-MHRW with LRU outperforms MHRW even without exact empirical measure due to it limited capacity. In most scenarios, HDT-MHRW with LRU leads to $10\%$ smaller TVD than the base MHRW with over $90\%$ memory reduction. The performance of HDT-MHRW with LRU is robust to the choice of $r$ in most cases. Due to space constraint, we defer more result of LRU scheme in plus-combined graph (over 100K nodes) to Appendix H.5.

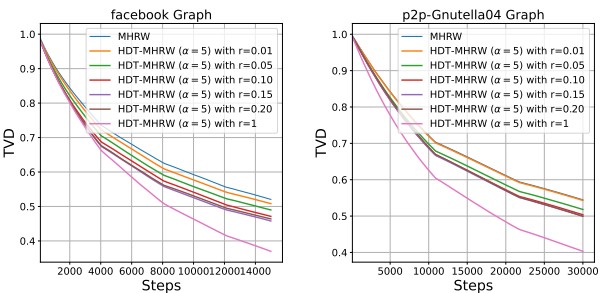

*Figure 4.* Performance of HDT-MHRW with LRU cache scheme.

## 5. Conclusion

In this paper, we propose a history-driven target (HDT) framework for MCMC sampling on general graphs. By embedding self-repellency in the target rather than the transition kernel, HDT maintains unbiased sampling with a lightweight design and provides an $O(1/\alpha)$ variance reduction without high computational cost or time-reversibility constraints from SRRW. Our theoretical analysis covers both reversible and non-reversible MCMC samplers, while empirical results on real-world graphs show robust performance gains and reduced computational overhead. To handle memory limitations, we introduce a Least Recently Used (LRU) cache scheme, enabling partial tracking of the empirical measure without loss in sampling efficiency. Future directions include more refined memory approximations for exponentially large configuration spaces and applications to high-dimensional statistical inference and network analysis.

## Acknowledgments and Disclosure of Funding

We thank anonymous reviewers for their constructive comments and Bohyung Han from Seoul National University for the valuable discussions. This work was supported in part by National Science Foundation under Grant Nos. CNS-2007423, IIS-2421484, and by Brain Pool program funded by the Ministry of Science and ICT through the National Research Foundation of Korea (No. RS-2024-00408610).

## Impact Statement

This paper presents work whose goal is to advance the field of Machine Learning. There are many potential societal consequences of our work, none which we feel must be specifically highlighted here.

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

## A. Matrix Form of Covariance Matrix (4)

The multivariate version of Brémaud (2013, eq. (6.33)) shows that

$$\lim_{t\to\infty} \frac{1}{t} \mathbb{E}\left[\left(\sum_{s=1}^{t} f(X_s) - \bar{f}\right)\left(\sum_{s=1}^{t} f(X_s) - \bar{f}\right)^T\right] = \text{Var}_{\boldsymbol{\mu}}(f(X_0)) + 2\sum_{i=2}^{|\mathcal{X}|} \frac{\lambda_i}{1-\lambda_i} \boldsymbol{f}^T \boldsymbol{D}_{\boldsymbol{\mu}} \boldsymbol{v}_i \boldsymbol{u}_i^T \boldsymbol{f}, \tag{16}$$

where $f(X) \in \mathbb{R}^d$, $\boldsymbol{f} = [f(i), f(j), \cdots]^T \in \mathbb{R}^{|\mathcal{X}|\times d}$, and $\text{Var}_{\boldsymbol{\mu}}(f(X_0)) = \boldsymbol{f}^T(\boldsymbol{D}_{\boldsymbol{\mu}} - \boldsymbol{\mu\mu}^T)\boldsymbol{f}$.

Replacing the function $f(\cdot)$ in (16) with our canonical vector $\boldsymbol{\delta}_{(\cdot)}$ defined in Section 2, we have

$$\begin{aligned}
\boldsymbol{V} &= \lim_{t\to\infty} \frac{1}{t} \mathbb{E}\left[\left(\sum_{s=1}^{t} \boldsymbol{\delta}_{X_s} - \boldsymbol{\mu}\right)\left(\sum_{s=1}^{t} \boldsymbol{\delta}_{X_s} - \boldsymbol{\mu}\right)^T\right] \\
&= \boldsymbol{D}_{\boldsymbol{\mu}} - \boldsymbol{\mu\mu}^T + 2\sum_{i=2}^{|\mathcal{X}|} \frac{\lambda_i}{1-\lambda_i} \boldsymbol{D}_{\boldsymbol{\mu}} \boldsymbol{v}_i \boldsymbol{u}_i^T \\
&= \boldsymbol{D}_{\boldsymbol{\mu}}\left(\sum_{i=1}^{|\mathcal{X}|} \boldsymbol{v}_i \boldsymbol{u}_i^T\right) - \boldsymbol{\mu\mu}^T + 2\boldsymbol{D}_{\boldsymbol{\mu}}\sum_{i=2}^{|\mathcal{X}|} \frac{\lambda_i}{1-\lambda_i} \boldsymbol{v}_i \boldsymbol{u}_i^T \\
&= \boldsymbol{D}_{\boldsymbol{\mu}}(\mathbf{1}\boldsymbol{\mu}^T) - \boldsymbol{\mu\mu}^T + \boldsymbol{D}_{\boldsymbol{\mu}}\sum_{i=2}^{|\mathcal{X}|} \frac{1+\lambda_i}{1-\lambda_i} \boldsymbol{v}_i \boldsymbol{u}_i^T \\
&= \sum_{i=2}^{|\mathcal{X}|} \frac{1+\lambda_i}{1-\lambda_i} \boldsymbol{D}_{\boldsymbol{\mu}} \boldsymbol{v}_i \boldsymbol{u}_i^T.
\end{aligned}$$

where the second equality is because $\boldsymbol{I} = [\boldsymbol{\delta}_i]_{i\in\mathcal{X}}^T$, and the third equality comes from the fact that $\boldsymbol{I} = \sum_{i=1}^{|\mathcal{X}|} \boldsymbol{v}_i \boldsymbol{u}_i^T$.

## B. Proof of Lemma 3.1

Following C2, we let $\pi_i[\mathbf{x}] = \frac{f(x_i, \mu_i)}{\sum_{j\in\mathcal{X}} f(x_j, \mu_j)}$ for some continuous differentiable function $f : \mathbb{R}_{>0} \times \mathbb{R}_{>0} \to \mathbb{R}_{>0} : (x, \mu) \to f(x, \mu)$. Then, C1 implies that for any scalars $c_1, c_2 > 0$,

$$\frac{f(c_1 x_i, c_2 \mu_i)}{\sum_{k\in\mathcal{X}} f(c_1 x_k, c_2 \mu_k)} = \frac{f(x_i, \mu_i)}{\sum_{k\in\mathcal{X}} f(x_k, \mu_k)}. \tag{17}$$

Now we show that function $f$ has a specific form

$$f(c_1 x, c_2 \mu) = g(c_1, c_2) f(x, \mu) \tag{18}$$

for any $x, \mu \in (0, 1)$, $c_1, c_2 > 0$ and some continuous differentiable function $g : (0, \infty)^2 \to \mathbb{R}_{>0} : (c_1, c_2) \to g(c_1, c_2)$.

*Proof of* (18). Rearranging (17) gives

$$f(c_1 x_i, c_2 \mu_i) = \frac{\sum_{k\in\mathcal{X}} f(c_1 x_k, c_2 \mu_k)}{\sum_{k\in\mathcal{X}} f(x_k, \mu_k)} \cdot f(x_i, \mu_i).$$

Then, with slight abuse of notation, we take $g(c_1, c_2, \mathbf{x}, \boldsymbol{\mu}) := \frac{\sum_{k\in\mathcal{X}} f(c_1 x_k, c_2 \mu_k)}{\sum_{k\in\mathcal{X}} f(x_k, \mu_k)}$. In other words,

$$f(c_1 x_i, c_2 \mu_i) = g(c_1, c_2, \mathbf{x}, \boldsymbol{\mu}) f(x_i, \mu_i) \tag{19}$$

for any $i \in \mathcal{X}$ and arbitrary choices $\mathbf{x}, \boldsymbol{\mu} \in \text{Int}(\Sigma)$. Then, we show that function $g(c_1, c_2, \mathbf{x}, \boldsymbol{\mu})$ is independent of parameters $\mathbf{x}, \boldsymbol{\mu}$, corresponding to (18).

Consider two pairs of probability vectors $(\mathbf{x}, \boldsymbol{\mu})$ and $(\tilde{\mathbf{x}}, \tilde{\boldsymbol{\mu}})$, whose $i$-th coordinate are identical, i.e., $x_i = \tilde{x}_i, \mu_i = \tilde{\mu}_i$. Then, $f(c_1 x_i, c_2 x_i) = f(c_1 \tilde{x}_i, c_2 \tilde{x}_i)$, and $f(x_i, x_i) = f(\tilde{x}_i, \tilde{x}_i)$. By (19), we have $g(c_1, c_2, \mathbf{x}, \boldsymbol{\mu}) = g(c_1, c_2, \tilde{\mathbf{x}}, \tilde{\boldsymbol{\mu}})$.

Similarly, we construct another pair $(\hat{\mathbf{x}}, \hat{\boldsymbol{\mu}})$ such that $\hat{x}_j = \tilde{x}_j$ and $\hat{\mu}_j = \tilde{\mu}_j$ for $j$-th coordinate (where $i \neq j$), then we have $g(c_1, c_2, \tilde{\mathbf{x}}, \tilde{\boldsymbol{\mu}}) = g(c_1, c_2, \hat{\mathbf{x}}, \hat{\boldsymbol{\mu}})$.

Therefore, the pair $(\tilde{\mathbf{x}}, \tilde{\boldsymbol{\mu}})$ serves as a 'bridge' to $(\mathbf{x}, \boldsymbol{\mu})$ and $(\hat{\mathbf{x}}, \hat{\boldsymbol{\mu}})$, leading to

$$g(c_1, c_2, \mathbf{x}, \boldsymbol{\mu}) = g(c_1, c_2, \hat{\mathbf{x}}, \hat{\boldsymbol{\mu}}). \tag{20}$$

This implies that for a given pair $(\mathbf{x}, \boldsymbol{\mu})$, we can always construct a distinct pair $(\hat{\mathbf{x}}, \hat{\boldsymbol{\mu}})$ such that (20) holds true, indicating that function $g$ must be identical for any pair $(\mathbf{x}, \boldsymbol{\mu})$. Hence, function $g$ cannot actually depend on $\mathbf{x}$ or $\boldsymbol{\mu}$; it must be a function of $(c_1, c_2)$ alone. $\qquad\square$

Next, we have the following proposition regarding the exact form of function $f(\cdot)$.[6]

**Proposition B.1.** *The function $f(x, \mu)$ satisfying (18) must be of the form*

$$f(x, \mu) = C x^{\rho_1} \mu^{\rho_2} \tag{21}$$

*for some positive constant $C$ and $\rho_1, \rho_2 \in \mathbb{R}$.*

*Proof of Proposition B.1.* Taking the logarithm of both sides of (18) gives:

$$\log f(c_1 x, c_2 \mu) = \log g(c_1, c_2) + \log f(x, \mu). \tag{22}$$

Define a function $F$ such that

$$F(\log x, \log \mu) = \log f(x, \mu).$$

Substituting function $F$ to (22) gives:

$$F(\log c_1 + \log x, \log c_2 + \log \mu) = \log g(c_1, c_2) + F(\log x, \log \mu).$$

Focusing on partial derivatives, let $c_2 = 1$, i.e., $\log c_2 = 0$, to isolate $x$. Then,

$$\frac{\partial}{\partial \log c_1} [F(\log c_1 + \log x, \log \mu) - F(\log x, \log \mu)] = \frac{\partial}{\partial \log c_1} \log g(c_1, 1).$$

As $c_1 \to 1$, $\frac{\partial}{\partial \log c_1} \log g(c_1, 1)$ approaches to some constant $\rho_1 \in \mathbb{R}$ irrelevant to $\mu$. Thus, the partial derivative of function $F(\log x, \log \mu)$ w.r.t $\log x$ is a constant and we have the form

$$F(\log x, \log \mu) = \rho_1 \log x + \text{ some function of } \log \mu.$$

Similarly, the partial derivative of $F(\log x, \log \mu)$ w.r.t $\log \mu$ is a constant $\rho_2 \in \mathbb{R}$ independent of $x$. Thus, we have

$$F(\log x, \log \mu) = \rho_1 \log x + \rho_2 \log \mu + b = \log f(x, \mu)$$

where $b$ is a constant. Therefore,

$$f(x, \mu) = e^b x^{\rho_1} \mu^{\rho_2},$$

which completes the proof.

$\qquad\square$

Following Proposition B.1, for $\rho_1, \rho_2 \in \mathbb{R}$, we have

$$\pi_i(\mathbf{x}) \propto x_i^{\rho_1} \mu_i^{\rho_2}, \tag{23}$$

---

[6]We admit that this result is likely to be well known but we were unable to locate reliable sources. Thus, we provide the proof for completeness.

where the constant $C$ in Proposition B.1 is absorbed into the normalizing constant. Then, based on the condition $\boldsymbol{\pi}[\boldsymbol{\mu}] = \boldsymbol{\mu}$ in C3, we have

$$\pi_i[\boldsymbol{\mu}] = \frac{\mu_i^{\rho_2} \mu_i^{\rho_1}}{\sum_k \mu_k^{\rho_2} \mu_k^{\rho_1}} = \mu_i.$$

Since this holds for any $i \in \mathcal{X}$, we have

$$\left(\frac{\mu_j}{\mu_i}\right)^{\rho_2 - 1} = \left(\frac{\mu_i}{\mu_j}\right)^{\rho_1} \iff \frac{\mu_i}{\mu_j} = \left(\frac{\mu_i}{\mu_j}\right)^{(1-\rho_2)/\rho_1},$$

thus $(1 - \rho_2)/\rho_1 = 1$, or equivalently, $\rho_1 + \rho_2 = 1$, and

$$\pi_i[\mathbf{x}] \propto \mu_i (x_i/\mu_i)^{1-\rho_2}.$$

Now, considering C4, we need to prioritize the node that is under-sampled, i.e., node $i$ with $x_i/\mu_i < 1$ should have a higher probability than $\mu_i$ such that $1 - \rho_2 \leq 1$. Thus, $\alpha \triangleq \rho_2 - 1 \geq 0$, and $\pi_i[\mathbf{x}] \propto \mu_i (x_i/\mu_i)^{-\alpha}$, which completes the proof.

## C. Implementation of Algorithm 1 For Advanced MCMC Samplers

### C.1. Multiple-try Metropolis algorithm with locally balanced weighs function (MTM)

We followed the MTM algorithm in (Chang et al., 2022), where weights function incorporated locally balanced function $h$ introduced by (Zanella, 2020). Specifically, the locally balanced weights function is defined as

$$w(j|i) = h\left(\frac{\mu_j Q_{ji}}{\mu_i Q_{ij}}\right) \quad \forall\, i, j \in \mathcal{X}, \tag{24}$$

where $Q$ is the random walk proposal. Thus, the corresponding locally balanced weights function of HDT-MTM is defined as

$$w_{\mathbf{x}_n}(j|i) = h\left(\frac{\pi_j[\mathbf{x}_n] Q_{ji}}{\pi_i[\mathbf{x}_n] Q_{ij}}\right) \quad \forall\, i, j \in \mathcal{X}, \tag{25}$$

where $\mathbf{x}$ is the empirical measure at step $n$. Typical choices of function $h(u)$ include $\sqrt{u}, \max(1, u), \min(1, u), u/(1 + u)$ and $1 + u$. In the simulation in Section 4, we choose $h$ as $h(u) = \sqrt{u}$, and the algorithm of HDT-MTM is described as below.

---

**Algorithm 2** HDT-MTM

---

**Input:** The number of candidates $K$, a weight function $w$ parameterized by $\mathbf{x}$ defined in (25), the number of iterations $T$.
**Initialization:** state $X_0 \in \mathcal{X}$, visit count $\tilde{x}(i) > 0, \forall i \in \mathcal{X}$.
**for** $n = 0$ **to** $T - 1$ **do**
    Step 1: Draw $\hat{Y}_1, \cdots, \hat{Y}_K$ u.a.r. from $\mathcal{N}(X_n)$, and compute $w_{\mathbf{x}_n}(\hat{Y}_k | X_n)$ for $k \in [K]$
    Step 2: Select $j \in [K]$ with probability proportional to $w_{\mathbf{x}_n}(\hat{Y}_j | X_n)$ and let $Y = \hat{Y}_j$
    Step 3: Sample $\hat{X}_1, \cdots, \hat{X}_{K-1}$ u.a.r. from $\mathcal{N}(Y)$ and compute $\alpha_{MTM}$ as

$$\alpha_{MTM} = \min\left\{1, \frac{\sum_{j \in [K]} w_{\mathbf{x}_n}(\hat{Y}_j | X_n)}{w_{\mathbf{x}_n}(X_n | Y) + \sum_{j \in [K-1]} w_{\mathbf{x}_n}(\hat{X}_j | Y)}\right\}$$

    Step 4: With probability $\alpha_{MTM}$, accept $Y$ and let $X_{n+1} = Y$; otherwise, $X_{n+1} = X_n$.
    Stpe 5: Update visit count $\tilde{x}(X_{n+1}) \leftarrow \tilde{x}(X_{n+1}) + 1$.
**end for**
**Output:** A set of samples $\{X_n\}_{n=1}^T$.

---

### C.2. Metropolis-Hastings algorithm with delayed acceptance (MHDA)

MHDA (Lee et al., 2012) is a nonreversible chain where it reduces the probability of the process from backtracking. Unlike standard MH always taking the next step once it accepted, MHDA avoid non-backtracking by re-proposing other neighboring

states uniformly with another acceptance probability whenever it is about to backtrack. This new acceptance probability is related to the base time-reversible chain $P$ with target $\mu$. It is proven to ensure the asymptotic variance of MHDA is no larger than that of MHRW. To apply our framework, we change the target distribution $\mu$ inside the first and the second acceptance probability by $\pi[\mathbf{x}]$. The detailed HDT-MHDA algorithm is described in Algorithm 3.

---

**Algorithm 3** HDT-MHDA

---

**Input:** The number of iterations $T$.
**Initialization:** state $X_0 \in \mathcal{X}$, last visit $Y_0 = X_0$, visit count $\tilde{x}(i) > 0, \forall i \in \mathcal{X}$.
**for** $n = 0$ **to** $T - 1$ **do**
    Step 1: Draw $k$ u.a.r. from $\mathcal{N}(X_n)$
    Step 2: Generate $p \sim U(0, 1)$
    **if** $p \leq \min\left\{1, \frac{\pi_k[\mathbf{x}_n]d(X_n)}{\pi_{X_n}[\mathbf{x}_n]d(k)}\right\}$ **then**

        **if** $Y_n = k$ and $d(X_n) > 1$ **then**
            Choose $r$ u.a.r from $\mathcal{N}(X_n) \setminus \{k\}$ and generate $q \sim U(0, 1)$
            **if** $q \leq \min\left\{1, \min\left\{1, \left(\frac{\pi_r[\mathbf{x}_n]d(X_n)}{\pi_{X_n}[\mathbf{x}_n]_{X_n}d(r)}\right)^2\right\} \max\left\{1, \left(\frac{\pi_{X_n}[\mathbf{x}_n]d(k)}{\pi_k[\mathbf{x}_n]d(X_n)}\right)^2\right\}\right\}$ **then**
                $X_{n+1} = r, Y_{n+1} = X_n$
            **else**
                $X_{n+1} = k, Y_{n+1} = X_n$
            **end if**
        **else**
            $X_{n+1} = k, Y_{n+1} = X_n$
        **end if**
    **else**
        $X_{n+1} = X_n, Y_{n+1} = Y_n$
    **end if**
    Step 3: Update visit count $\tilde{x}(X_{n+1}) \leftarrow \tilde{x}(X_{n+1}) + 1$.
**end for**
**Output:** A set of samples $\{X_n\}_{n=1}^T$.

---

### C.3. 2-cycle MCMC

HDT-2-cycle MCMC alternates between two reversible Markov chains ($P_1$ and $P_2$) targeting the history-dependent distribution $\pi[\mathbf{x}]$. At each iteration $n$, if $n$ is even, the next state $X_{n+1}$ is generated using $P_1$; if odd, $P_2$ is used. Both chains adaptively target $\pi[\mathbf{x}_n]$ via the updated empirical measure $\mathbf{x}_n$ at each step. After sampling the state $X_{n+1}$, the visit count $\tilde{x}(X_{n+1})$ is incremented by 1, which will guide exploration by penalizing over-visited states, as detailed in Algorithm 4.

---

**Algorithm 4** HDT-2-cycle MCMC

---

**Input:** Two reversible chain $P_1$ and $P_2$ with respect to target $\mu$, the number of iterations $T$.
**Initialization:** state $X_0 \in \mathcal{X}$, visit count $\tilde{x}(i) > 0, \forall i \in \mathcal{X}$.
**for** $n = 0$ **to** $T - 1$ **do**
    Step 1: (Run chain $P$)
    **if** n%2 == 0 **then**
        Generate sample $X_{n+1}$ from chain $P_1(X_n, \cdot)$ with target $\pi[\mathbf{x}_n]$
    **else**
        Generate sample $X_{n+1}$ from chain $P_2(X_n, \cdot)$ with target $\pi[\mathbf{x}_n]$
    **end if**
    Step 2: Update visit count $\tilde{x}(X_{n+1}) \leftarrow \tilde{x}(X_{n+1}) + 1$.
**end for**

---

## D. Proof of Lemma 3.2

The proof consists of three parts: We first show that

$$V(\mathbf{x}) = \sum_{i \in \mathcal{X}} \mu_i (x_i/\mu_i)^{-\alpha} \tag{26}$$

is the Lyapunov function of the ODE

$$\dot{\mathbf{x}} = \boldsymbol{\pi}[\mathbf{x}] - \mathbf{x}. \tag{27}$$

Then, we prove that $\boldsymbol{\mu}$ is the unique fixed point of the ODEs.

Last, with LaSalle's invariance principle (Khalil, 2002, Corollary 4.2), we can prove that $\boldsymbol{\mu}$ is globally asymptotically stable.

**Part I.** To prove (26) is the Lyapunov function, we take partial derivative of $V(\mathbf{x})$ w.r.t $t$ such that

$$
\begin{aligned}
\frac{\partial V(\mathbf{x})}{\partial t} &= \sum_{i \in \mathcal{X}} \frac{\partial V(\mathbf{x})}{\partial x_i} \dot{x}_i \\
&= -\alpha \sum_{i \in \mathcal{X}} \left( \frac{x_i}{\mu_i} \right)^{-\alpha-1} \left( \frac{\mu_i (x_i/\mu_i)^{-\alpha}}{\sum_{k \in \mathcal{X}} \mu_k (x_k/\mu_k)^{-\alpha}} - x_i \right) \\
&= -\alpha \sum_{i \in \mathcal{X}} \left[ \frac{\mu_i (x_i/\mu_i)^{-2\alpha-1}}{\sum_{k \in \mathcal{X}} \mu_k (x_k/\mu_k)^{-\alpha}} + \mu_i \left( \frac{x_i}{\mu_i} \right)^{-\alpha} \right] \\
&= \frac{-\alpha}{V(\mathbf{x})} \left[ \sum_{i \in \mathcal{X}} \mu_i (x_i/\mu_i)^{-2\alpha-1} - \left( \sum_{i \in \mathcal{X}} \mu_i \left( \frac{x_i}{\mu_i} \right)^{-\alpha} \right)^2 \right] \\
&= \frac{-\alpha}{V(\mathbf{x})} \left[ \sum_{i \in \mathcal{X}} x_i (x_i/\mu_i)^{-2\alpha-2} - \left( \sum_{i \in \mathcal{X}} x_i \left( \frac{x_i}{\mu_i} \right)^{-\alpha-1} \right)^2 \right] \leq 0,
\end{aligned}
$$

where the fourth equality comes from the definition of $V(\mathbf{x})$ in (26), and the last inequality is from Cauchy-Schwartz inequality by rewriting

$$
\left( \sum_{i \in \mathcal{X}} x_i \left( \frac{x_i}{\mu_i} \right)^{-\alpha-1} \right)^2 = \left( \sum_{i \in \mathcal{X}} x_i^{1/2} \cdot x_i^{1/2} \left( \frac{x_i}{\mu_i} \right)^{-\alpha-1} \right)^2 \leq \underbrace{\left( \sum_{i \in \mathcal{X}} x_i \right)}_{=1} \left( \sum_{i \in \mathcal{X}} x_i (x_i/\mu_i)^{-2\alpha-2} \right),
$$

and its equality holds only when $x_i^{1/2} = C x_i^{1/2} (x_i/\mu_i)^{-\alpha-1}$ for some constant $C$ and any $i \in \mathcal{X}$. Or equivalently, $\mathbf{x} = \boldsymbol{\mu}$. Then, $\dot{V}(\mathbf{x}) \leq 0$ and the equality holds only when $\mathbf{x} = \boldsymbol{\mu}$, which is the equilibrium of the ODE (27). Thus, $V(\mathbf{x})$ in the form of (26) is the Lyapunov function of the ODE (27).

**Part II.** Then, we show that $\boldsymbol{\mu}$ is the unique fixed point of $\boldsymbol{\pi}[\mathbf{x}]$ in C3. Let $\mathbf{x}^*$ be the fixed point of $\boldsymbol{\pi}[\mathbf{x}]$, i.e.,

$$\pi_i[\mathbf{x}^*] = \frac{\mu_i (x_i^*/\mu_i)^{-\alpha}}{\sum_{k \in \mathcal{X}} \mu_k (x_k^*/\mu_k)^{-\alpha}} = x_i^*, \forall i \in \mathcal{X}.$$

Rearranging and combining terms on both sides of the equation gives

$$(x_i^*/\mu_i)^{-\alpha-1} = \sum_{k \in \mathcal{X}} \mu_k (x_k^*/\mu_k)^{-\alpha}.$$

Since the RHS is invariant to index $i$, we have

$$(x_i^*/\mu_i)^{-\alpha-1} = (x_j^*/\mu_j)^{-\alpha-1}, \forall i, j \in \mathcal{X}.$$

Since $\alpha \geq 0$, we have $x_i^*/\mu_i = x_j^*/\mu_j$ for all $i, j \in \mathcal{X}$, implying that $x_i^* = C\mu_i$ for some scalar $C$ and any $i \in \mathcal{X}$, and thus $\mathbf{x}^* = \boldsymbol{\mu}$ because $\mathbf{x}^*, \boldsymbol{\mu} \in Int(\Sigma)$.

**Part III.** We first provide the LaSalle's invariance principle below for self-contained purpose.

**Theorem D.1** (LaSalle's invariance principle (Khalil, 2002)). *Let $\mathbf{x}^*$ be an equilibrium point for the ODE $\dot{\mathbf{x}} = f(\mathbf{x})$. Let $V : dom f \to \mathbb{R}$ be a continuously differentiable, radially unbounded function, positive definite function such that $\dot{V}(\mathbf{x}) \leq 0$ for all $\mathbf{x} \in dom f$. Let $S = \{\mathbf{x} \in dom f | \dot{V}(\mathbf{x}) = 0\}$ and suppose that no solution can stay identically in $S$ other than the trivial solution $\mathbf{x}(t) \equiv \mathbf{x}^*$. Then, $\mathbf{x}^*$ is globally asymptotically stable.*

To apply Theorem D.1, we let $f(\mathbf{x}) \triangleq \boldsymbol{\pi}[\mathbf{x}] - \mathbf{x}$ such that $dom f = Int(\Sigma)$. The set $S = \{\boldsymbol{\mu}\}$. By construction, $V(\mathbf{x}) \geq 0$ and $V(\mathbf{x}) \to \infty$ as $\mathbf{x} \to \partial\Sigma$ (the boundary of the probability simplex, where at least one entry $x_i$ of the empirical measure $\mathbf{x}$ becomes zero). Along with the design of $\boldsymbol{\pi}[\mathbf{x}]$ in (6) following that $\boldsymbol{\mu}$ is the unique fixed point, we prove that $\boldsymbol{\mu}$ is globally asymptotically stable for the ODE (27).

# E. Proof of Theorem 3.3

We leverage the stochastic approximation theories from Delyon (2000) and Fort (2015) for asymptotic analysis within our HDT-MCMC framework regarding the iteration of the empirical measure $\mathbf{x}_n$. We obtain almost sure convergence and CLT results in Appendix E.1 and E.2, as was done similarly for SRRW in Doshi et al. (2023). However, we adopt the asymptotic analysis to non-reversible Markov chains as detailed in Appendix E.3, to which SRRW cannot accommodate.

To start with, we first introduce the following theorem for almost sure and CLT results of the stochastic approximation.

**Theorem E.1** ((Delyon, 2000) Theorem 15, (Fort, 2015) Theorem 3.2). *Consider the following stochastic approximation:*

$$\mathbf{x}_{n+1} = \mathbf{x}_n + \gamma_{n+1} H(\mathbf{x}_n, X_{n+1}), \tag{28}$$

*where $\{X_n\}_{n \geq 0}$ is driven by a Markov chain $\boldsymbol{P}[\mathbf{x}]$, parameterized by $\mathbf{x} \in \mathbb{R}^d$, with stationary distribution $\boldsymbol{\pi}[\mathbf{x}]$, and function $H : \mathbb{R}^d \times \mathcal{X} \to \mathbb{R}^d : (\mathbf{x}, X) \to H(\mathbf{x}, X)$. Moreover, the mean field $h(\mathbf{x}) = \mathbb{E}_{i \sim \boldsymbol{\pi}[\mathbf{x}]}[H(\mathbf{x}, i)]$. For the following conditions:*

*(B1) Function $h : \mathbb{R}^d \to \mathbb{R}^d$ is continuous on $\mathcal{X}$, there exists a non-negative differentiable, radially unbounded function $V$ such that $\langle \nabla V(\mathbf{x}), h(\mathbf{x}) \rangle \leq 0, \forall \mathbf{x} \in \mathbb{R}^d$ and the set $\mathcal{S} = \{\mathbf{x} \mid \langle \nabla V(\mathbf{x}), h(\mathbf{x}) \rangle = 0\}$ is such that $V(\mathcal{S})$ has empty interior. There exists a compact set $\mathcal{K} \subset \mathbb{R}^d$ such that $\langle \nabla V(\mathbf{x}), h(\mathbf{x}) \rangle < 0$ if $\mathbf{x} \notin \mathcal{K}$;*

*(B2) For every $\mathbf{x}$, there exists a function $m_{\mathbf{x}}(i)$ such that for every $i \in \mathcal{X}$,*

$$m_{\mathbf{x}}(i) - (\boldsymbol{P}[\mathbf{x}]m_{\mathbf{x}})(i) = H(\mathbf{x}, i) - h(\mathbf{x}). \tag{29}$$

*For any compact set $\mathcal{C} \subset \mathbb{R}^d$,*

$$\sup_{\mathbf{x} \in \mathcal{C}, i \in \mathcal{X}} \|H(\mathbf{x}, i)\|_2 + \|m_{\mathbf{x}}(i)\|_2 < \infty. \tag{30}$$

*There exists a continuous function $\phi_{\mathcal{C}}$, $\phi_{\mathcal{C}}(0) = 0$, such that for any $\mathbf{x}, \mathbf{x}' \in \mathcal{C}$,*

$$\sup_{i \in \mathcal{X}} \|(\boldsymbol{P}[\mathbf{x}]m_{\mathbf{x}})(i) - (\boldsymbol{P}[\mathbf{x}']m_{\mathbf{x}'})(i)\|_2 \leq \phi_{\mathcal{C}} \left( \|\mathbf{x} - \mathbf{x}'\|_2 \right). \tag{31}$$

*(B3) The step size follows $\gamma_n \geq 0, \sum_{n \geq 1} \gamma_n = \infty, \sum_{n \geq 1} \gamma_n^2 < \infty$ and $\sum_{n \geq 1} |\gamma_{n+1} - \gamma_n| < \infty$.*

*(B4) $\sup_n \|\mathbf{x}_n\|_2 < \infty$ almost surely.*

*(B5) Function $h$ is continuous, differentiable in some neighborhood of $\mathbf{x}^* \in \mathcal{S}$, and matrix $\nabla h(\mathbf{x})|_{x=\boldsymbol{\mu}}$ (written as $\nabla h(\mathbf{x}^*)$ for brevity) has all its eigenvalues with negative real parts.*

*If (B1) - (B4) are satisfied, then almost surely,*

$$\limsup_n \liminf_{\mathbf{x}^* \in \mathcal{S}} \|\mathbf{x}_n - \mathbf{x}^*\|_2 = 0. \tag{32}$$

*If additionally (B5) is satisfied, then condition on $\mathbf{x}_n \to \mathbf{x}^*$, where $\mathbf{x}^* \in \mathcal{S}$, we have*

$$\sqrt{n}(\mathbf{x}_n - \mathbf{x}^*) \xrightarrow[dist.]{n \to \infty} N(\mathbf{0}, \boldsymbol{V}),$$

*where*

$$V = \int_0^\infty e^{(\nabla h(\mathbf{x}^*)+\boldsymbol{I}/2)t}\boldsymbol{U}e^{(\nabla h(\mathbf{x}^*)+\boldsymbol{I}/2)^T t}dt, \tag{33}$$

*and*

$$\boldsymbol{U} = \lim_{t\to\infty}\frac{1}{t}\mathbb{E}\left[\left(\sum_{s=1}^t (H(\mathbf{x}^*, X_s) - h(\mathbf{x}^*))\right)\left(\sum_{s=1}^t (H(\mathbf{x}^*, X_s) - h(\mathbf{x}^*))\right)^T\right]. \tag{34}$$

$\square$

We interpret the iteration of empirical measure $\mathbf{x}_n$ of Algorithm 1 here as an instance of stochastic approximation.

$$\mathbf{x}_{n+1} = \mathbf{x}_n + \frac{1}{n+2}\underbrace{(\boldsymbol{\delta}_{X_{n+1}} - \mathbf{x}_n)}_{\triangleq H(\mathbf{x}_n, X_{n+1})} = \mathbf{x}_n - \frac{1}{n+2}\underbrace{(\boldsymbol{\pi}[\mathbf{x}_n] - \mathbf{x}_n)}_{\triangleq h(\mathbf{x}_n)} + \frac{1}{n+2}(\boldsymbol{\delta}_{X_{n+1}} - \boldsymbol{\pi}[\mathbf{x}_n]). \tag{35}$$

### E.1. Ergodicity

We examine conditions (B1) - (B4) to obtain the almost sure convergence. (B1) stems from the stability of the related ODE, which has been proved in Appendix D Part III. Specifically, the Lyapunov function $V$ is of the form in (26), and $\mathcal{S} = \mathcal{K} = \{\boldsymbol{\mu}\}$ contains a singleton $\boldsymbol{\mu}$, which is the unique fixed point of the ODE, or equivalently, $h(\boldsymbol{\mu}) = \boldsymbol{\pi}[\boldsymbol{\mu}] - \boldsymbol{\mu} = 0$. (B3) is automatically satisfied since $\gamma_n = 1/(n+1)$ as in (35). (B4) is guaranteed since $\mathbf{x} \in \text{Int}(\Sigma)$ so that $\|\mathbf{x}_n\|_2 \le 1$ almost surely for any $n \ge 0$.

Now, we focus on (B2). In our Algorithm 1, replacing the target of the base MCMC sampler from $\boldsymbol{\mu}$ to $\boldsymbol{\pi}[\mathbf{x}]$ results in an ergodic transition matrix $\boldsymbol{P}[\mathbf{x}]$ parameterized by $\mathbf{x}$, which is $\boldsymbol{\pi}[\mathbf{x}]$-invariant. For a function $m : \mathcal{X} \to \mathbb{R}^d$, we define the operator

$$(\boldsymbol{P}[\mathbf{x}]m)(i) \triangleq \sum_{j \in \mathcal{X}} P_{ij}[\mathbf{x}]m(j).$$

For the function $H(\mathbf{x}, \cdot) : \mathcal{X} \to \mathbb{R}^{|\mathcal{X}|}$, there always exists a corresponding function $m_{\mathbf{x}}(\cdot) : \mathcal{X} \to \mathbb{R}^{|\mathcal{X}|}$ that satisfies the Poisson equation (29). According to Hu et al. (2024b, Appendix C), the solution $m_{\mathbf{x}}(i)$ to the Poisson equation (29) is of the form

$$m_{\mathbf{x}}(i) = \sum_{j \in \mathcal{X}} \left(\boldsymbol{I} - \boldsymbol{P}[\mathbf{x}] + \mathbf{1}\boldsymbol{\pi}[\mathbf{x}]^T\right)_{ij}^{-1} (H(\mathbf{x}, j) - h(\mathbf{x})), \tag{36}$$

where $(\boldsymbol{I} - \boldsymbol{P}[\mathbf{x}] + \mathbf{1}\boldsymbol{\pi}[\mathbf{x}]^T)^{-1}$ is the fundamental matrix and it is always well-defined whenever $\boldsymbol{P}[\mathbf{x}]$ is an ergodic Markov chain with stationary distribution $\boldsymbol{\pi}[\mathbf{x}]$ (Brémaud, 2013, Chapter 6.3.1). Because $\mathbf{x}, \boldsymbol{\pi}[\mathbf{x}] \in \text{Int}(\Sigma)$ the probability simplex and $\boldsymbol{\delta}_{(\cdot)}$ is the canonical vector, we have $\|H(\mathbf{x}, i)\|_2 = \|\boldsymbol{\delta}_i - \mathbf{x}\| \le 2$, as well as $\|h(\mathbf{x})\|_2 = \|\boldsymbol{\pi}[\mathbf{x}] - \mathbf{x}\| \le 2$, we have $\|m_{\mathbf{x}}(i)\|_2 \le 2\sum_{j \in \mathcal{X}} \left(\boldsymbol{I} - \boldsymbol{P}[\mathbf{x}] + \mathbf{1}\boldsymbol{\pi}[\mathbf{x}]^T\right)_{ij}^{-1} < \infty$ for all $i \in \mathcal{X}$. Therefore, (30) is satisfied.

Next, note that the MCMC sampler for the target $\boldsymbol{\mu}$ is associated with a transition kernel $\boldsymbol{P}$, exhibiting continuity in relation to $\boldsymbol{\mu}$. Since $\boldsymbol{\pi}[\mathbf{x}]$ in the form of (6) is continuous in $\mathbf{x}$, we have $\boldsymbol{P}[\mathbf{x}]$ continuous in its target $\boldsymbol{\pi}[\mathbf{x}]$, and in turn continuous in $\mathbf{x} \in \text{Int}(\Sigma)$. As a consequence, (36) is continuous in $\mathbf{x}$ such that for every compact subset $\mathcal{C}$ of probability simplex $\Sigma$, $(\boldsymbol{P}[\mathbf{x}]m_{\mathbf{x}})(i)$ is continuous in $\mathbf{x} \in \mathcal{C}$ for any $i \in \mathcal{X}$, and thus Lipschitz in $\mathcal{C}$. Therefore, (31) holds. As a result, (B2) is satisfied, and (32) shows $\mathbf{x}_n \to \boldsymbol{\mu}$ as $n \to \infty$ almost surely.

### E.2. CLT

To obtain the CLT result, we examine (B5). Note that $\boldsymbol{\pi}[\mathbf{x}]$ in the form of (6) is continuous and differentiable in $\mathbf{x} \in \text{Int}(\Sigma)$. In addition, we take the partial derivative of $h(\mathbf{x})$ w.r.t $\mathbf{x}$ and obtain the following: After some algebraic computations, for $j \ne i$,

$$\frac{\partial}{\partial x_j}h_i(x) = \frac{\partial}{\partial x_j}\left[\frac{\mu_i(x_i/\mu_i)^{-\alpha}}{\sum_{k \in \mathcal{X}} \mu_k(x_k/\mu_k)^{-\alpha}} - x_i\right] = \frac{\alpha\pi_i[\mathbf{x}]\pi_j[\mathbf{x}]}{x_j}.$$

Similarly, when $j = i$,

$$\frac{\partial}{\partial x_i}h_i(x) = -\frac{\alpha\pi_i[\mathbf{x}](1 - \pi_i[\mathbf{x}])}{x_i} - 1.$$

Setting $\mathbf{x} = \boldsymbol{\mu}$ in above partial derivatives gives

$$\frac{\partial}{\partial x_j} h_i(x) = \alpha\mu_i, \quad \frac{\partial}{\partial x_i} h_i(x) = \alpha\mu_i - (\alpha + 1),$$

or equivalently, $\nabla h(\boldsymbol{\mu}) = \alpha\boldsymbol{\mu}\mathbf{1}^T - (\alpha + 1)\boldsymbol{I}$, which has all eigenvalues no larger than $-1$. Thus, (B5) is satisfied.

Now, we have the CLT result from Theorem E.1. We translate the form of $\boldsymbol{V}, \boldsymbol{U}$ from (33) and (34) to our settings. (3) shows that

$$\boldsymbol{U} = \lim_{t\to\infty} \frac{1}{t}\mathbb{E}\left[\left(\sum_{s=1}^{t}(\boldsymbol{\delta}_{X_s} - \boldsymbol{\mu})\right)\left(\sum_{s=1}^{t}(\boldsymbol{\delta}_{X_s} - \boldsymbol{\mu})\right)^T\right] = \boldsymbol{V}^{\text{base}}.$$

Before proceeding with the expression of matrix $\boldsymbol{V}^{\text{HDT}}(\alpha)$, we first provide a result: $\mathbf{1}^T\boldsymbol{U} = \mathbf{0}^T$ by considering the above expression of matrix $\boldsymbol{U}$ and $\mathbf{1}^T\boldsymbol{\delta}_{X_s} - \mathbf{1}^T\boldsymbol{\mu} = 0$. Then,

$$\begin{aligned}
\boldsymbol{V}^{\text{HDT}}(\alpha) &= \int_0^\infty e^{(\nabla h(\boldsymbol{\mu})+\boldsymbol{I}/2)t}\boldsymbol{U}e^{(\nabla h(\boldsymbol{\mu})+\boldsymbol{I}/2)^T t}dt \\
&= \int_0^\infty e^{(\alpha\boldsymbol{\mu}\mathbf{1}^T - (\alpha+1/2)\boldsymbol{I}/2)t}\boldsymbol{U}e^{(\alpha\mathbf{1}\boldsymbol{\mu}^T - (\alpha+1/2)\boldsymbol{I}/2)^T t}dt \\
&= \int_0^\infty \left(\sum_{i=2}^{|\mathcal{X}|} e^{-(\alpha+1/2)t}\boldsymbol{u}_i\boldsymbol{v}_i^T + e^{-t/2}\boldsymbol{\mu}\mathbf{1}^T\right)\boldsymbol{U}\left(\sum_{i=2}^{|\mathcal{X}|} e^{-(\alpha+1/2)t}\boldsymbol{u}_i\boldsymbol{v}_i^T + e^{-t/2}\boldsymbol{\mu}\mathbf{1}^T\right)^T dt \\
&= \int_0^\infty \left(\sum_{i=2}^{|\mathcal{X}|} e^{-(\alpha+1/2)t}\boldsymbol{u}_i\boldsymbol{v}_i^T\right)\boldsymbol{U}\left(\sum_{i=2}^{|\mathcal{X}|} e^{-(\alpha+1/2)t}\boldsymbol{u}_i\boldsymbol{v}_i^T\right)^T dt \\
&= \sum_{i=2}^{|\mathcal{X}|}\sum_{j=2}^{|\mathcal{X}|}\int_0^\infty \left(e^{-(2\alpha+1)t}dt\right)\left(\boldsymbol{u}_i\boldsymbol{v}_i^T\boldsymbol{U}\boldsymbol{v}_j\boldsymbol{u}_j^T\right) \\
&= \frac{1}{2\alpha+1}\left(\sum_{i=2}^{|\mathcal{X}|}\boldsymbol{u}_i\boldsymbol{v}_i^T\right)\boldsymbol{U}\left(\sum_{j=2}^{|\mathcal{X}|}\boldsymbol{v}_j\boldsymbol{u}_j^T\right) \\
&= \frac{1}{2\alpha+1}(\boldsymbol{I} - \boldsymbol{\mu}\mathbf{1}^T)\boldsymbol{U}(\boldsymbol{I} - \mathbf{1}\boldsymbol{\mu}^T) \\
&= \frac{1}{2\alpha+1}\boldsymbol{U} \\
&= \frac{1}{2\alpha+1}\boldsymbol{V}^{\text{base}},
\end{aligned}$$

where the third equality comes from the eigendecomposition of the matrix exponential form. The fourth and the second last equalities stem from the fact $\mathbf{1}^T\boldsymbol{U} = \mathbf{0}$. The third last equality comes from the fact that $\boldsymbol{I} = \sum_{i=1}^{|\mathcal{X}|}\boldsymbol{u}_i\boldsymbol{v}_i^T = \boldsymbol{\mu}\mathbf{1}^T + \sum_{i=2}^{|\mathcal{X}|}\boldsymbol{u}_i\boldsymbol{v}_i^T$ and $\boldsymbol{I}^T = \mathbf{1}\boldsymbol{\mu}^T + \sum_{i=2}^{|\mathcal{X}|}\boldsymbol{v}_i\boldsymbol{u}_i^T = \boldsymbol{I}$.

### E.3. Adaptation to Non-Reversible Markov Chains

For non-reversible Markov chains that directly modify the transition probabilities such as Suwa & Todo (2010); Chen & Hwang (2013); Bierkens (2016); Thin et al. (2020), they typically add some target-dependent mass to the standard MH kernel $\boldsymbol{P}$ with target $\boldsymbol{\mu}$, and let it satisfy the 'skew detailed balance' condition, e.g., Thin et al. (2020), eq. (3)). For example, Bierkens (2016) introduces a vorticity matrix $\boldsymbol{\Gamma}_{\boldsymbol{\mu}}$, which always exists, to break the reversibility while ensuring that the chain retains an arbitrary target distribution $\boldsymbol{\mu} \in \text{Int}(\Sigma)$. When applied in our Algorithm 1, for each given empirical measure $\mathbf{x}$, there will be a specific $\boldsymbol{\Gamma}_{\boldsymbol{\pi}[\mathbf{x}]}$, still maintaining the target distribution as $\boldsymbol{\pi}[\mathbf{x}]$. This vorticity matrix $\boldsymbol{\Gamma}_{\boldsymbol{\pi}[\mathbf{x}]}$ always exists for a given $\mathbf{x}$ because $\boldsymbol{\pi}[\mathbf{x}]$ is simply a new target distribution. Therefore, in this case, for any given $\mathbf{x}$, the resulting transition kernel $\boldsymbol{P}[\mathbf{x}]$ emerging from the non-reversible Markov chain is still ergodic and $\boldsymbol{\pi}[\mathbf{x}]$-invariant. We note that in the proof of ergodicity and CLT, *we never require the time reversibility* of the underlying Markov chain $\boldsymbol{P}[\mathbf{x}]$ for a given $\mathbf{x} \in \text{Int}(\Sigma)$. For

the proof, the sole requirement is the continuity of the kernel $P$, determined by its target $\mu$, from the non-reversible MCMC algorithms. Consequently, this line of work can be directly covered by the theoretical analysis in Appendix E.1 and E.2.

However, the above argument does not work for non-reversible Markov chains with augmented state spaces, like MHDA and 2-cycle Markov chain, which incorporate an additional variable to induce directional bias in transitions (Lee et al., 2012; Maire et al., 2014; Andrieu & Livingstone, 2021), rather than acting on the original state space $\mathcal{X}$. Specifically, their trajectory is usually defined as $\{(X_n, Y_n)\}_{n \geq 0}$, where the sample $X_n \in \mathcal{X}$ is defined in the original state space $\mathcal{X}$, and $Y_n \in \mathcal{Y}$ is defined on some state space, such as $\mathcal{Y} = \{0, 1\}$ for a 2-cycle Markov chain such that $Y_n \in \{0, 1\}$ to indicate which Markov chain to use in each iteration, and $\mathcal{Y} = \mathcal{X}$ for MHDA such that $Y_n = X_{n-1}$, which represents the most recent sample.

Recall the stochastic approximation (28) requires $\{X_n\}$ to be controlled Markovian dynamics. That is, $X_{n+1}$ is drawn from a transition kernel $P_{(X_n, \cdot)}[\mathbf{x}_n]$ parameterized by empirical measure $\mathbf{x}_n$. If the chain instead operates on an augmented space $(\mathcal{X}, \mathcal{Y})$, then $\{X_n\}$ in isolation cannot be viewed as a controlled Markov chain, since $X_{n+1}$ depends not only on $X_n, \mathbf{x}_n$, but also on the auxiliary state $Y_n \in \mathcal{Y}$. More concretely, mixing these spaces (using a transition matrix on $(\mathcal{X}, \mathcal{Y})$ but a function $H(\mathbf{x}, \cdot)$ only defined on $\mathcal{X}$) causes a dimensional mismatch. For example, in (36), the solution $m_{\mathbf{x}}(i)$ is not well defined because $P[\mathbf{x}]$ is specified on $(\mathcal{X}, \mathcal{Y})$, while $H(\mathbf{x}, \cdot)$ is only on $\mathcal{X}$. On the other hand, a natural solution to the above mismatch is to record visit frequencies of the augmented state in the empirical measure update (35). This resolves a dimensional mismatch but inflates the dimension of $\mathbf{x}$. For example, this would increase the dimension of $\mathbf{x}$ from $|\mathcal{X}|$ to $|\mathcal{X}|^2$ for the MHDA and to $2|\mathcal{X}|$ for the 2-cycle Markov chain. This is especially problematic for large state space $\mathcal{X}$.

In the following, we provide a trick to accommodate the proofs in Appendix E.1 and E.2 to the non-reversible Markov chains on the augmented state space without modifying the empirical measure $\mathbf{x}$, i.e., the empirical measure is still defined on the original state space $\mathcal{X}$.

Given a target $\mu$, we denote by $\hat{\mu}$ the stationary distribution defined on the augmented state space $\mathcal{X} \times \mathcal{Y}$, and admits the marginal stationary distribution $\mu_i = \sum_{j \in \mathcal{Y}} \hat{\mu}_{(i,j)}$ because these non-reversible Markov chains are designed to achieve the same target distribution $\mu$ on the original state space $\mathcal{X}$, e.g., Lee et al. (2012, Theorem 6), Maire et al. (2014, Proposition 13). Thus, when adopting these algorithms under our HDT-MCMC framework, for a given $\mathbf{x} \in \text{Int}(\Sigma)$, we alter their target distribution by $\boldsymbol{\pi}[\mathbf{x}]$, and let $\hat{\boldsymbol{\pi}}[\mathbf{x}]$ be the joint distribution of the augmented states with $\pi_i[\mathbf{x}] = \sum_{j \in \mathcal{Y}} \hat{\pi}_{(i,j)}[\mathbf{x}]$. Then, we define another function $\Phi : \mathbb{R}^{|\mathcal{X}|} \times \mathcal{X} \times \mathcal{Y} \to \mathbb{R}^{|\mathcal{X}|}$ such that $\Phi(\mathbf{x}, i, j) = H(\mathbf{x}, i) = \boldsymbol{\delta}_i - \mathbf{x}$. Then, the $\mathbf{x}_n$ update (35) becomes

$$\mathbf{x}_{n+1} = \mathbf{x}_n + \frac{1}{n+2} \Phi(\mathbf{x}_n, X_{n+1}, Y_{n+1}). \tag{37}$$

For any $\mathbf{x} \in \text{Int}(\Sigma)$, we have

$$\phi(\mathbf{x}) \triangleq \mathbb{E}_{(i,j) \sim \hat{\boldsymbol{\pi}}[\mathbf{x}]} \Phi(\mathbf{x}, i, j) = \sum_{i \in \mathcal{X}, j \in \mathcal{Y}} H(\mathbf{x}, i) \hat{\pi}_{(i,j)}[\mathbf{x}] = \sum_{i \in \mathcal{X}} H(\mathbf{x}, i) \sum_{j \in \mathcal{Y}} \hat{\pi}_{(i,j)}[\mathbf{x}] = \sum_{i \in \mathcal{X}} H(\mathbf{x}, i) \pi_i[\mathbf{x}] = h(\mathbf{x}).$$

Thus, the mean field $\phi(\cdot)$ is identical to $h(\cdot)$. Therefore, instead of analyzing (35), the analyses in Appendix E.1 and E.2 now directly carry over to the iteration (37), where we replace the function $H$ by $\Phi$ and its mean field $h$ by $\phi$.

Moreover, the matrix $U$ is given by

$$U = \lim_{t \to \infty} \frac{1}{t} \mathbb{E} \left\{ \left[ \sum_{s=1}^{t} \Phi(\theta^*, Z_s) \right] \left[ \sum_{s=1}^{t} \Phi(\theta^*, Z_s) \right]^T \right\}$$

$$= \lim_{t \to \infty} \frac{1}{t} \mathbb{E} \left\{ \left[ \sum_{s=1}^{t} H(\boldsymbol{\mu}, X_s) \right] \left[ \sum_{s=1}^{t} H(\boldsymbol{\mu}, X_s) \right]^T \right\},$$

which is exactly the definition of the asymptotic covariance for non-reversible Markov chains, as studied in Lee et al. (2012); Maire et al. (2014); Andrieu & Livingstone (2021). Therefore, we maintain the same expression of $V$ from Appendix E.2 for non-reversible Markov chains on augmented state spaces. This completes the proof.

*Remark* E.1. A caveat here is that we no longer have the alternative form for the matrix $V$ as in (4) because the transition kernel for the non-reversible Markov chain is defined on the augmented state space $\mathcal{X} \times \mathcal{Y}$, while the asymptotic covariance here is only on the original state space $\mathcal{X}$. This is the reason that in Theorem 3.3, we refer to the original definition of the asymptotic variance in (3) rather than (4).

## F. Proof of Theorem 3.5

Due to the fact that $\mathbf{x}_n - \boldsymbol{\mu} = \frac{1}{n}\sum_{s=1}^{n}(\boldsymbol{\delta}_{X_s} - \boldsymbol{\mu})$, we first rewrite

$$\sqrt{B}(\mathbf{x}_{T^{\text{HDT}}(B)} - \boldsymbol{\mu}) = \sqrt{B} \cdot \frac{1}{T^{\text{HDT}}(B)} \sum_{s=1}^{T^{\text{HDT}}(B)} (\boldsymbol{\delta}_{X_s} - \boldsymbol{\mu}) = \sqrt{\frac{B}{T^{\text{HDT}}(B)}} \cdot \underbrace{\frac{1}{\sqrt{T^{\text{HDT}}(B)}} \sum_{s=1}^{T^{\text{HDT}}(B)} (\boldsymbol{\delta}_{X_s} - \boldsymbol{\mu})}_{\text{Term A}}. \tag{38}$$

To deal with Term A, we state the random-change-of-time theory as follows.

**Theorem F.1** ((Billingsley, 2013) Theorem 14.4)**.** *Given a sequence of random variable* $\theta_1, \theta_2, \cdots$ *with partial sum* $S_n \triangleq \sum_{s=1}^{n} \theta_s$ *such that*

$$\frac{1}{\sqrt{n}} S_n \xrightarrow[dist.]{n \to \infty} N(\mathbf{0}, \boldsymbol{V}).$$

*Let* $n_l$ *be some positive random variable taking integer value such that* $\theta_{n_l}$ *is on the same space as* $\theta_n$. *In addition, for some deterministic sequence* $\{c_l\}_{l \geq 0}$, *which goes to infinity,* $n_l/c_l \to c$ *for a positive constant c. Then,*

$$\frac{1}{\sqrt{n_l}} S_{n_l} \xrightarrow[dist.]{l \to \infty} N(\mathbf{0}, \boldsymbol{V}). \qquad \square$$

By Theorem 3.3(b), we have $\sqrt{n}(\mathbf{x}_n - \boldsymbol{\mu}) \xrightarrow[dist.]{n \to \infty} N(\mathbf{0}, \boldsymbol{V}(\alpha))$, and $\sqrt{n}(\mathbf{x}_n - \boldsymbol{\mu}) = \frac{1}{\sqrt{n}}\sum_{s=1}^{n}(\boldsymbol{\delta}_{X_s} - \boldsymbol{\mu})$. We have a deterministic sequence $\{1, 2, \cdots, \lfloor B \rfloor\}$, where $\lfloor \cdot \rfloor$ is the floor function, and the condition $B/T^{\text{HDT}}(B) \to C^{\text{HDT}}$ in Theorem 3.5. Thus, in view of Theorem F.1, we have

$$\frac{1}{\sqrt{T^{\text{HDT}}(B)}} \sum_{s=1}^{T^{\text{HDT}}(B)} (\boldsymbol{\delta}_{X_s} - \boldsymbol{\mu}) \xrightarrow[dist.]{B \to \infty} N(\mathbf{0}, \boldsymbol{V}). \tag{39}$$

Now, we state the Slutsky's theorem as follows.

**Theorem F.2** (Slutsky's theorem (Ash & Doléans-Dade, 2000))**.** *Let* $\{A_n\}$ *and* $\{B_n\}$ *be the sequence of random variables. If* $A_n \xrightarrow[dist.]{n \to \infty} A$ *for some distribution A and* $B_n \to b$ *almost surely as* $n \to \infty$ *for a non-zero constant b, then*

$$A_n/B_n \xrightarrow[dist.]{n \to \infty} A/b. \qquad \square$$

Following (39), $B/T^{\text{HDT}}(B) \to C^{\text{HDT}}$ almost surely, and Theorem F.2, as $B \to \infty$, (38) converges weakly to $N(\mathbf{0}, C^{\text{HDT}} \boldsymbol{V}(\alpha))$. We can follow the same procedures for SRRW, which completes the proof.

## G. Proof of Lemma 3.6

We first briefly explain the cost of our HDT-MCMC and SRRW per sample. At each step, our HDT-MCMC computes the proposal distribution $Q_{ij}$ and $Q_{ji}$ with 2 costs. Thus, we have $a_i = 2c$ and $C^{\text{HDT}} = 2c$.

On the other hand, in SRRW, normalizing $K_{ij}[\mathbf{x}]$ over all $j \in \overline{\mathcal{N}}(i)$ costs $2c|\overline{\mathcal{N}}(i)|$, since each neighbor's acceptance ratio must be calculated. Based on the definition in (12), we derive the inequality for $B/T^{\text{SRRW}}(B)$:

$$\frac{\sum_{n=1}^{k+1} 2c|\overline{\mathcal{N}}(X_n)|}{k} \geq \frac{B}{T^{\text{SRRW}}(B)} \geq \frac{\sum_{n=1}^{k} 2c|\overline{\mathcal{N}}(X_n)|}{k}. \tag{40}$$

Taking $B \to \infty$ is equivalent to taking $k \to \infty$, so the RHS of (40) becomes

$$\lim_{k \to \infty} \frac{2c}{k} \sum_{n=1}^{k} |\overline{\mathcal{N}}(X_n)| = 2c\mathbb{E}_{i \sim \boldsymbol{\mu}}[|\overline{\mathcal{N}}(i)|],$$

where the equality comes from the ergodicity in Theorem 3.3(a). Similarly, we can rewrite the LHS of (40) as

$$\frac{\sum_{n=1}^{k} 2c|\overline{\mathcal{N}(X_n)}|}{k} + \frac{2c|\overline{\mathcal{N}(X_{n+1})}|}{k}.$$

Since $c|\overline{\mathcal{N}(X_{n+1})}| < \infty$, the LHS of (40) still converges to $2c\mathbb{E}_{i\sim\boldsymbol{\mu}}[|\overline{\mathcal{N}(i)}|]$ as $k \to \infty$. Hence, $C^{\mathrm{SRRW}} = 2c\mathbb{E}_{i\sim\boldsymbol{\mu}}[|\overline{\mathcal{N}(i)}|]$, and

$$C^{\mathrm{SRRW}}/C^{\mathrm{HDT}} = \mathbb{E}_{i\sim\boldsymbol{\mu}}[|\overline{\mathcal{N}(i)}|]. \tag{41}$$

Note that the 'expanded' neighborhood size $|\mathcal{N}(i)| \geq 2$ because the graph is connected so that each state has at least one neighbor plus itself, thus $\mathbb{E}_{i\sim\boldsymbol{\mu}}[|\overline{\mathcal{N}(i)}|] \geq 2$.

From (5), (10) and by noting the eigenvalue $\lambda_i \leq 1$, we show that

$$
\begin{aligned}
\boldsymbol{V}^{\mathrm{SRRW}}(\alpha) &= \sum_{i=2}^{|\mathcal{X}|} \frac{1}{2\alpha(\lambda_i + 1) + 1} \cdot \frac{1 + \lambda_i}{1 - \lambda_i} \boldsymbol{u}_i \boldsymbol{u}_i^T \\
&\succeq \frac{1}{4\alpha + 1} \sum_{i=2}^{|\mathcal{X}|} \frac{1 + \lambda_i}{1 - \lambda_i} \boldsymbol{u}_i \boldsymbol{u}_i^T \\
&= \frac{2\alpha + 1}{4\alpha + 1} \boldsymbol{V}^{\mathrm{HDT}}(\alpha) \\
&\succeq \frac{1}{2} \boldsymbol{V}^{\mathrm{HDT}}(\alpha).
\end{aligned}
\tag{42}
$$

where the second inequality (by Loewner ordering) is because $\lambda_i \leq 1$ such that $\frac{1}{2\alpha(\lambda_i+1)+1} - \frac{1}{4\alpha+1} \geq 0$ and $\sum_{i=2}^{|\mathcal{X}|}(\frac{1}{2\alpha(\lambda_i+1)+1} - \frac{1}{4\alpha+1})\frac{1+\lambda_i}{1-\lambda_i}\boldsymbol{u}_i\boldsymbol{u}_i^T$ is positive semi-definite using the definition of Loewner ordering, i.e., for any non-zero vector $\boldsymbol{z}$,

$$\boldsymbol{z}^T \left( \sum_{i=2}^{|\mathcal{X}|} \left( \frac{1}{2\alpha(\lambda_i + 1) + 1} - \frac{1}{4\alpha + 1} \right) \frac{1 + \lambda_i}{1 - \lambda_i} \boldsymbol{u}_i \boldsymbol{u}_i^T \right) \boldsymbol{z} = \sum_{i=2}^{|\mathcal{X}|} \left( \frac{1}{2\alpha(\lambda_i + 1) + 1} - \frac{1}{4\alpha + 1} \right) \frac{1 + \lambda_i}{1 - \lambda_i} (\boldsymbol{z}^T \boldsymbol{u}_i)^2 \geq 0.$$

The last inequality comes from $\frac{2\alpha+1}{4\alpha+1} \geq \frac{1}{2}$ such that $\frac{2\alpha+1}{4\alpha+1}\boldsymbol{V}^{\mathrm{HDT}}(\alpha) - \frac{1}{2}\boldsymbol{V}^{\mathrm{HDT}}(\alpha)$ is positive semi-definite (note that $\boldsymbol{V}^{\mathrm{HDT}}(\alpha)$ is positive semi-definite by its definition in (10)).

Then, by Theorem 3.5, we have

$$
\begin{aligned}
C^{\mathrm{SRRW}}\boldsymbol{V}^{\mathrm{SRRW}}(\alpha) &= C^{\mathrm{HDT}}\mathbb{E}[|\overline{\mathcal{N}(i)}|]\boldsymbol{V}^{\mathrm{SRRW}}(\alpha) \\
&\succeq C^{\mathrm{HDT}}\boldsymbol{V}^{\mathrm{HDT}}(\alpha) \cdot \frac{\mathbb{E}[|\overline{\mathcal{N}(i)}|]}{2},
\end{aligned}
$$

where the first equality comes from (41), and the second inequality is from (42). This completes the proof.

## H. Additional Simulation

### H.1. Simulation setting for graph sampling

*Table 2.* Summary of graph datasets.

| Name | # of nodes. | # of edges | Average degree |
|---|---|---|---|
| WikiVote | 889 | 2,914 | 6.55 |
| Facebook | 4039 | 88,234 | 43.69 |
| p2p-Gnutella08 | 6301 | 20,777 | 6.59 |
| p2p-Gnutella04 | 10876 | 39,994 | 7.35 |

Table 2 shows the detailed summary of the graph used in our experiments. We used different numbers of iterations for each graph $T = \{3000, 15000, 15000, 3000\}$, respectively. In addition to TVD, we also simulate normalized root mean

square error (NRMSE) under HDT-MCMC framework in graph sampling, where the NRMSE of the MCMC estimator $\psi_n(f) \triangleq \frac{1}{n} \sum_{s=1}^{n} f(s)$ for some test function $f : \mathcal{X} \to \mathbb{R}$ with the ground truth $\bar{\psi} \triangleq \mathbb{E}_{\mu}[f]$ is defined as

$$\mathrm{NRMSE}(\psi_n, \bar{\psi}) = \sqrt{\mathbb{E}[(\psi_n(f) - \bar{\psi})^2]}/\bar{\psi}$$

.

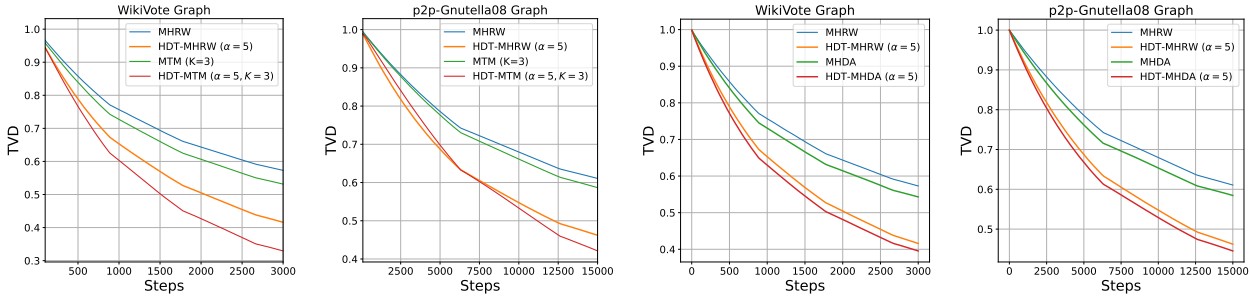

*Figure 5.* TVD improvement over HDT framework. The upper row for nonreversible chain; TVD improvement over HDT-MTM with locally balanced weights function where MHRW is a special case when $K = 1$.

### H.2. More results on HDT-MCMC in graph sampling

In this section, we present more simulation results on the algorithm that utilizes HDT-MCMC framework for graph sampling in other graph. To demonstrate the impact of HDT-MCMC, we implement HDT-version of both reversible and nonreversible chain. For instance, we implement HDT-MTM as a reversible chain, and HDT-MHDA, HDT-2-cycle as non-reversible chains. A detailed implementation can be found in Appendix C. We use TVD and NRMSE as the metrics, and set our test function as $f : \mathcal{X} \to \{0, 1\}$. This form of test function can be interpreted as the label of each node. Estimating the density of the node label is crucial, especially in online social networks.

For each graph, we first randomly assign nodes with label 1 with some probability $p$ and 0 otherwise. Namely, Let $\mathcal{S}_1$ be the set of the node that is assigned to label 1, i.e. $\mathcal{S}_1 = \{i \in \mathcal{X} | f(i) = 1\}$ We aim to estimate the true proportion of label 1 using graph sampling. The assigned probability $p$ is set as 0.3. Figure 6 include the result of MHRW, MTM, MHDA and their corresponding HDT-version. We observe that HDT version algorithms consistently outperform the base version across different graphs. We also notice that HDT version algorithms have better performance than SRRW, as shown in Facebook graph and p2p-Gnutella08 graph.

*Remark* H.1. We conjecture that TVD metric reflects the exploration speed of the random walker, as it equally weights contributions from visited and unvisited states. SRRW achieves superior performance by adjusting its transition kernel to reduce self-transition probabilities. In contrast, HDT-MCMC's use of the Metropolis-Hastings (MH) algorithm may increase self-transition probabilities compared to SRRW. However, for graph sampling, even partial graph exploration with an unbiased MCMC estimator can accurately approximate the true value effectively, explaining its improved NRMSE performance.

Figure 5 shows the TVD result of HDT-MCMC framework over the graph dataset. The improvement in TVD is also consistent among all the graph, showing the benefits of utilizing HDT-MCMC framework can convergence faster than the base chain.

We also implemented HDT version of 2-cycle MCMC as an another nonreversible chain that be benefits from this works. In particular, we used MTM and MH as two reversible chains in 2-cycle MCMC while its HDT version is constructed by replacing MTM and MH with their HDT versions. Figure 7 shows the result where HDT version 2-cycle MCMC outperforms the base 2-cycle chain among all the graph dataset.

*Figure 6.* NRMSE Comparison for HDT framework.

## H.3. Different Initializations on State $X_0$ and Fake Visit Counts $\mathbf{x}_0$

Here, we conduct experiments under different setups to demonstrate the robustness of HDT-MCMC across several aspects. In addition to Facebook graph presented in the main body, we further evaluate the robustness of HDT-MCMC to the choice of initial state $X_0$ using p2p-Gnutella04 graph. The results in Table 3 and Table 4 show both TVD and NRMSE maintain consistent performance across different initializations.

*Table 3.* Mean (std error) TVD at $n = 30,000$ step with different $X_0$ initializations on p2p-Gnutella04. All std error values are in units of $10^{-4}$.

|  | Low-Deg Group | High-Deg Group |
| --- | --- | --- |
| MHRW | 0.545 (1.738) | 0.545 (1.704) |
| HDT-MHRW | 0.403 (0.748) | 0.403 (0.765) |
| MTM | 0.514 (1.546) | 0.514 (1.556) |
| HDT-MTM | 0.328 (0.999) | 0.328 (1.035) |
| MHDA | 0.522 (1.616) | 0.521 (1.549) |
| HDT-MHDA | 0.388 (0.708) | 0.388 (0.747) |

*Table 4.* Mean NRMSE at $n = 30,000$ step with different $X_0$ initializations on p2p-Gnutella04.

|  | Low-Deg Group | High-Deg Group |
| --- | --- | --- |
| MHRW | 0.030 | 0.031 |
| HDT-MHRW | 0.013 | 0.013 |
| MTM | 0.025 | 0.025 |
| HDT-MTM | 0.012 | 0.013 |
| MHDA | 0.028 | 0.029 |
| HDT-MHDA | 0.013 | 0.012 |

Next, we consider experiments in which the fake visit count $\mathbf{x}_0$ is initialized from

- 'Deg' scenario: Fake visit counts set by node degrees, heavily favoring high-degree nodes.

- 'Non-unif' scenario: Fake visit counts randomly drawn from a Dirichlet distribution with default concentration parameter of $0.5$.

- 'Unif' scenario: Same initial count for all nodes.

Here, we use Facebook graph and p2p-Gnutella04 graph in our simulation and show TVD in Tables 5 - 6, and NRMSE in Tables 7 - 8, both with confidence intervals $95\%$. Similar to the conclusions from the results on different initialization on state $X_0$, our HDT framework is robust to different fake visit counts, and our HDT version still showcases improvements over their baseline counterparts in both TVD and NRMSE metrics.

*Table 5.* Mean (std error) TVD at $n = 15,000$ with different fake visit counts $\mathbf{x}_0$ on Facebook. All std error values are in units of $10^{-3}$

|  | Deg | Non-unif | Unif |
|---|---|---|---|
| MHRW | 0.520 (2.256) | 0.520 (2.256) | 0.520 (2.256) |
| HDT-MHRW | 0.371 (1.246) | 0.371 (1.281) | 0.371 (1.246) |
| MTM | 0.487 (2.128) | 0.487 (2.128) | 0.487 (2.128) |
| HDT-MTM | 0.288 (1.751) | 0.285 (1.456) | 0.285 (1.496) |
| MHDA | 0.513 (2.175) | 0.513 (2.1.75) | 0.513 (2.175) |
| HDT-MHDA | 0.365 (1.281) | 0.365 (1.246) | 0.366 (1.258) |

*Table 6.* Mean (std error) TVD at $n = 30,000$ with different fake visit counts $\mathbf{x}_0$ on p2p-Gnutella04. All std error values are in units of $10^{-4}$.

|  | Deg | Non-unif | Unif |
|---|---|---|---|
| MHRW | 0.545 (1.621) | 0.545 (1.621) | 0.545 (1.621) |
| HDT-MHRW | 0.403 (0.744) | 0.403 (0.742) | 0.403 (0.745) |
| MTM | 0.514 (1.587) | 0.514 (1.587) | 0.514 (1.587) |
| HDT-MTM | 0.325 (1.031) | 0.330 (0.987) | 0.328 (1.069) |
| MHDA | 0.522 (1.573) | 0.522 (1.573) | 0.522 (1.573) |
| HDT-MHDA | 0.388 (0.757) | 0.388 (0.702) | 0.388 (0.733) |

*Table 7.* Mean NRMSE at $n = 15,000$ step with different fake visit counts $\mathbf{x}_0$ on Facebook.

|  | Deg | Non-unif | Unif |
|---|---|---|---|
| MHRW | 0.079 | 0.079 | 0.079 |
| HDT-MHRW | 0.028 | 0.028 | 0.028 |
| MTM | 0.056 | 0.056 | 0.056 |
| HDT-MTM | 0.098 | 0.047 | 0.062 |
| MHDA | 0.068 | 0.068 | 0.068 |
| HDT-MHDA | 0.027 | 0.028 | 0.027 |

*Table 8.* Mean NRMSE at $n = 15,000$ step with different fake visit counts $\mathbf{x}_0$ on p2p-Gnutella04.

|  | Deg | Non-unif | Unif |
|---|---|---|---|
| MHRW | 0.030 | 0.030 | 0.030 |
| HDT-MHRW | 0.013 | 0.013 | 0.013 |
| MTM | 0.023 | 0.023 | 0.023 |
| HDT-MTM | 0.013 | 0.012 | 0.012 |
| MHDA | 0.028 | 0.028 | 0.028 |
| HDT-MHDA | 0.013 | 0.012 | 0.013 |

## H.4. Effect of $\alpha$ on Convergence

To illustrate the impact of $\alpha$, we run HDT-MHRW under $\alpha \in \{0, 1, 2, 5, 10\}$, with $\alpha = 0$ being the MHRW. We observed in Figure 8 that the performance of average TVD is asymptotically improved with larger $\alpha$, supporting the theoretical result in Theorem 3.3 where larger $\alpha$ leads to smaller covariance.

## H.5. LRU on Larger graph

Here, we test our LRU with HDT-MHRW on WikiVote graph and p2p-Gnutella graph, and the result is shown in Figure 9. We also observed LRU with 80% memory reduction can still outperforms standard MH. Moreover, in p2p-Gnutella08 graph, we notice that the performance of LRU design is robust to the choice of the memory capacity. A comparable performance

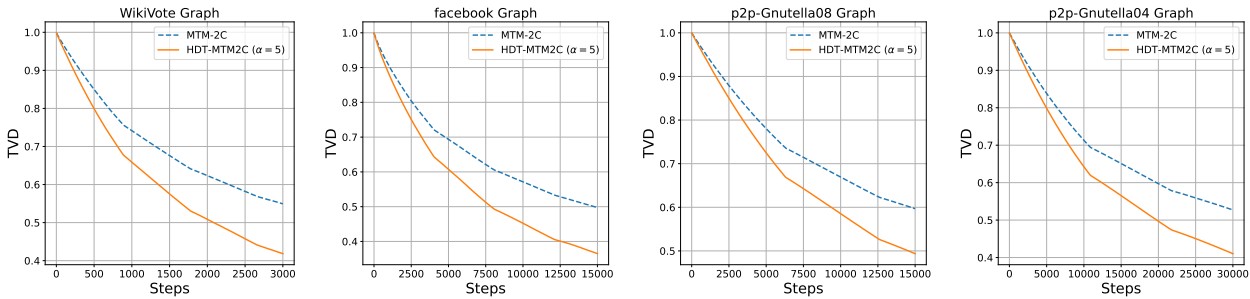

*Figure 7.* TVD Comparison for HDT-2cyc.

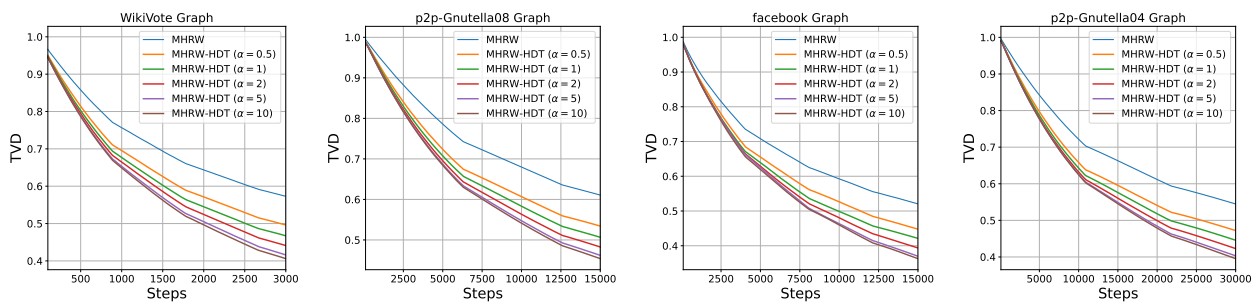

*Figure 8.* Simulation of HDT-MHRW with different choices of $\alpha$ where MHRW can be viewed as the special case when $\alpha = 0$.

when $r$ is chosen within 5 to 20.

Furthermore, we test our design using the ego-Gplus graph (Leskovec & Krevl, 2014), which contains about 100K nodes, and implemented $r = 0.01$ and $r = 0.1$ in LRU. For simulation setting, We run 50 independent trials with 300,000 steps and set one-third of the number of steps as burn-in period before collecting samples. Figure 11 shows the result. We observed a similar robustness behavior of the choice of LRU capacity with respect to average TVD. The performance of using $r = 0.01$ is comparable to using $r = 0.1$ where both cases outperforms MHRW. This findings open a possibility that we may only requires an approximated empirical measure to implement HDT-MCMC while remaining fast convergence. Furthermore, the corresponding theoretical result of that can be one possible future work.

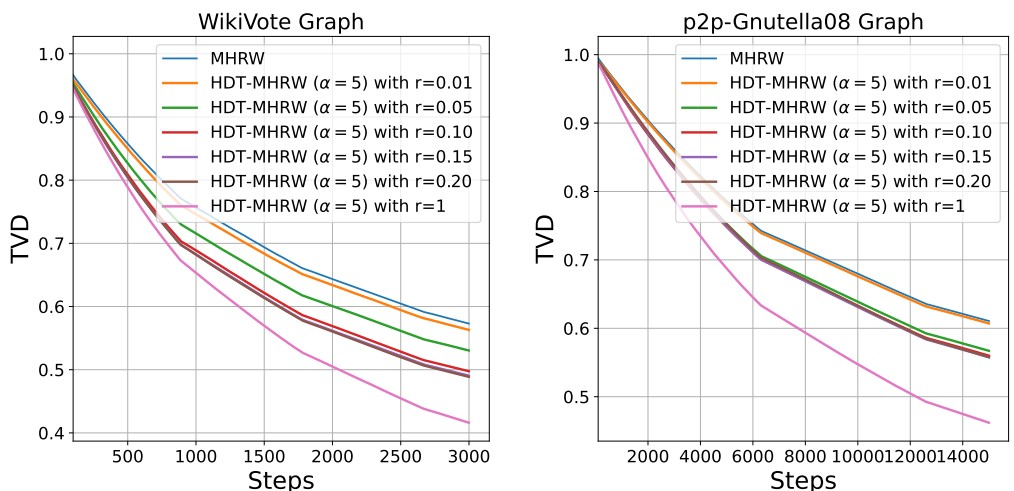

*Figure 9.* Simulation of incorporating LRU in HDT-MHRW framework to reduce memory issues.

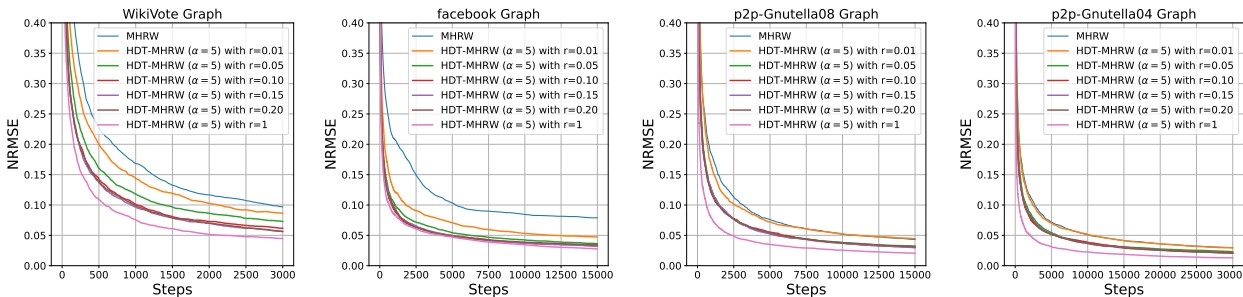

*Figure 10.* NRMSE result of HDT-MHRW framework with LRU

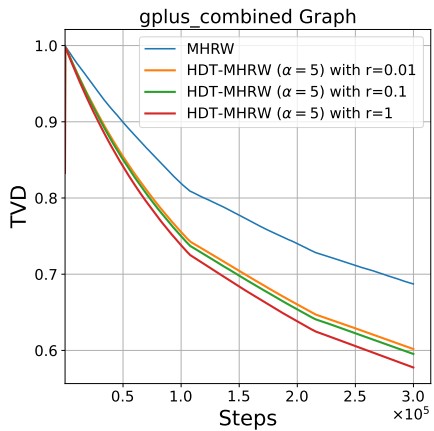

*Figure 11.* Simulation of the HDT-MHRW with different LRU capacity for graph sampling on gplus-combined graph.

### H.6. Additional Experiments on Non-Uniform Target

Since HDT-MCMC can generally be applied to any target distribution $\boldsymbol{\mu}$, we conducts additional experiments on non-uniform target with HDT-MCMC. In particular, we conduct MHRW, MTM and MHDA with their HDT improvements over AS-733 graph (with $6474$ nodes and $13895$ edges) (Leskovec et al., 2005) and Whois graph (with $7476$ nodes and $56.9K$ edges) (Mahadevan et al., 2006), and set the target distribution proportional to the degree distribution, i.e. $\mu_i \propto d_i$ with $\alpha = 1$ in HDT. Similar to the simulation settings in our manuscript, we compare HDT-MCMC to baseline algorithms through TVD (with $95\%$ confidence interval in shaded region) and NRMSE metrics. Since the target distribution is no longer uniform, we adopt importance sampling (Doshi et al., 2023) to obtain an unbiased estimator and then compute NRMSE. The following experimental results are obtained through $100$ independent runs.

We first show the comparison between HDT-MCMC and baseline algorithms using the TVD metric in Figure 12, where HDT outperforms its counterpart in both AS-733 and Whois graphs. In addition, for two baseline Then, we show the performance ordering of HDT-MHRW in terms of different $\alpha$ values. As illustrated in Figure 13, a larger $\alpha$ leads to a smaller TVD, aligning with our theoretical findings in our Theorem 3.3. We also test the performance of HDT-MCMC with heuristic LRU cache scheme in Figure 14. In the Whois graph, HDT-MHRW combined with LRU results in a $10\%$ smaller TVD compared to the base MHRW, with over a $90\%$ decrease in memory usage. On the other hand, on the AS-733 graph, the LRU caching scheme yields only modest benefits, achieving a $5\%$ TVD reduction with approximately $50\%$ less memory consumption (with $r = 0.5$). In Figure 15, we compare the performance of HDT-MHRW and SRRW the under the same computational budget. Since SRRW always requires computing the transition probability around its neighbor at each iteration, it induces a large computation during sampling. In contrast, HDT-MHRW only queries one node per iteration, making it more lightweight. In the experiment, we set $\alpha = 1$ and the total steps as 30000 without a burn-in period. We also observe HDT-MCMC achieves better performance in terms of TVD metric given any fixed budget. Moreover, the performance gap between SRRW and HDT-MHRW is larger on the Whois graph, where average degree of is approximately 15 and is larger than average degree of AS 733, around $4$. This also aligns with our Lemma 3.6 where the HDT framework

provide more efficiency in dense graph under limited budget.

In the simulation of estimating the density of the node label, we retain the same test functions as in Appendix H.2 in our submission. However, since the target distribution is no longer uniform, we have to use importance reweighting to adjust the weight of each sample. In Figure 16, we show that in the long run, HDT-MCMC still outperforms its baseline counterpart.

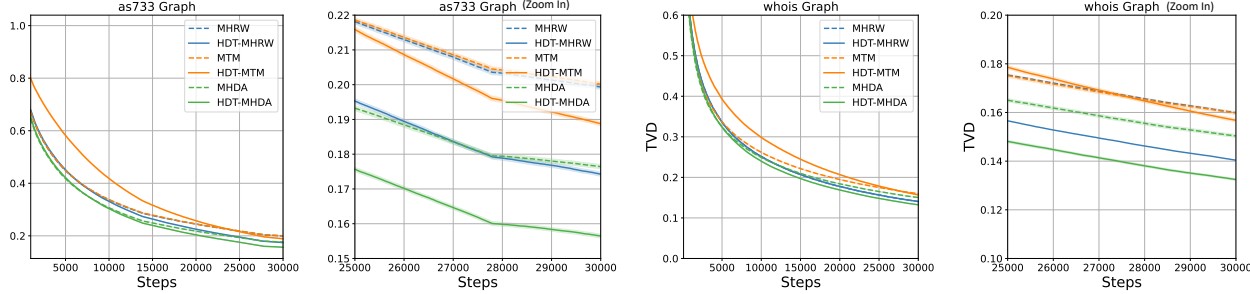

*Figure 12.* Comparison of HDT-MCMC with base chain MHRW, MTM, and MHDA via TVD metric in AS-733 and Whois graphs. The plots in the right column is the zoom in version of those in the left column.

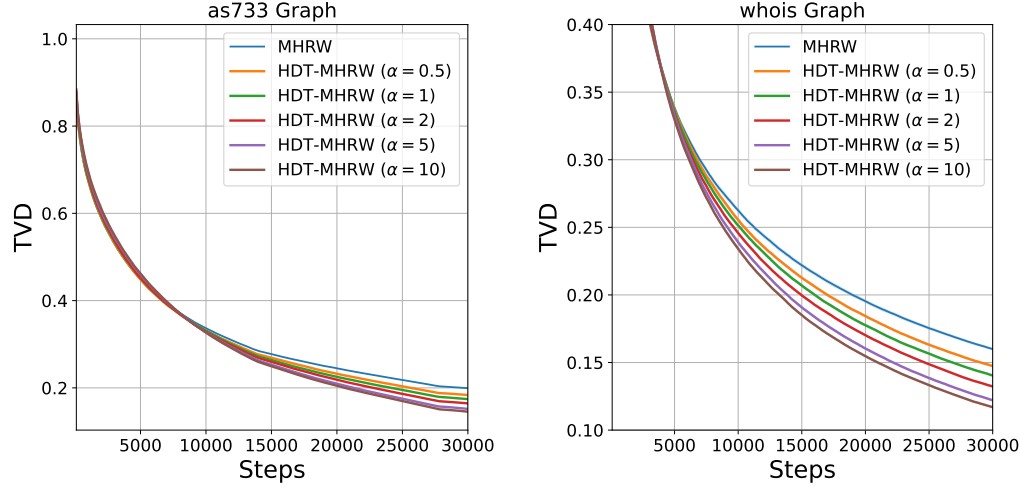

*Figure 13.* Simulation of HDT-MHRW with different choices of $\alpha$ where MHRW can be viewed as the special case when $\alpha = 0$.

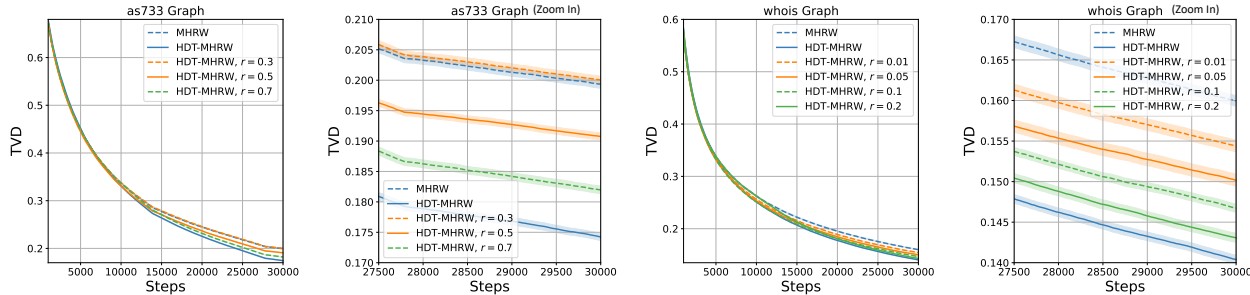

*Figure 14.* Performance of HDT-MHRW with LRU cache scheme.

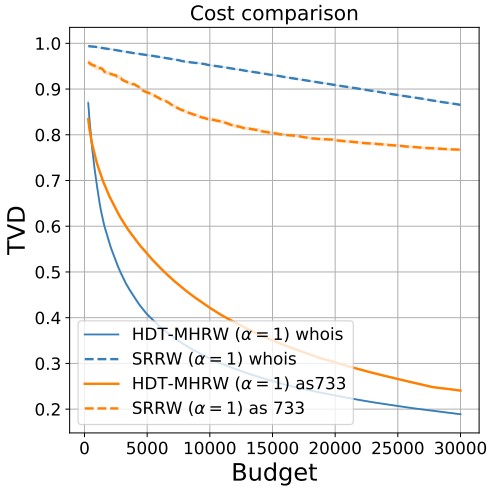

*Figure 15.* TVD comparison between HDT-MHRW and SRRW under budget constraints.

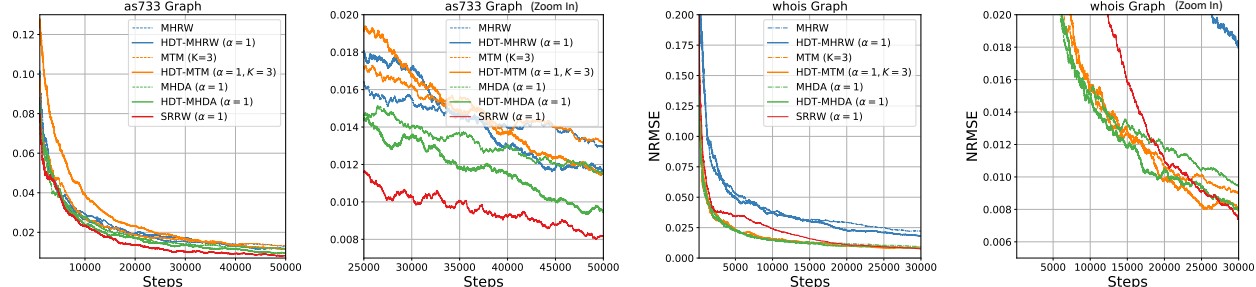

*Figure 16.* Comparison of HDT-MCMC (with thicker solid curves) to base chain MHRW, MTM, and MHDA (with thinner dash-dot curves) via NRMSE metric in AS-733 and Whois graphs. The plots in the right column is the zoom in version of those in the left column.

