# OpenReview forum: "Beyond Self-Repellent Kernels: History-Driven Target Towards Efficient Nonlinear MCMC on General Graphs"
_ICML.cc/2025/Conference — ICML 2025 oral_

### Official Review · Reviewer_nUqH · 2025-03-10

**Overall Recommendation:** 4

**Summary:**

The paper introduces a new "history-driven target" (HDT) framework to improve Markov Chain Monte Carlo (MCMC) sampling on graphs. HDT modifies the target distribution based on the history of visited states, improving efficiency. This helps overcome the computational limitations of the recently-proposed class of self-repellent random walks (SRRW), which in contrast uses a nonlinear transition kernel that scales poorly with graph size. Unlike SRRW, HDT is compatible with both reversible and non-reversible MCMC samplers. The authors make strong theoretical contributions and test their claims with experiments. They introduce a heuristic scheme that they call the “least recently used (LRU) cache” to further reduce storage requirements.

## Update after rebuttal:
The authors have clarified the downstream benefits of their algorithm, the impact of graph properties on its effectiveness, and performance with a non-uniform target measure. I'm satisfied with the responses and have raised my score.

**Claims And Evidence:**

The claims are supported by theoretical and empirical evidence. See below.

**Essential References Not Discussed:**

The references seem reasonable.

**Experimental Designs Or Analyses:**

See above. The experiments do indeed back up the authors’ central claims, but they do not convincingly demonstrate downstream benefits for ML or scaling to very large graphs. Only a limited number of graphs are considered (2 in the main body, and a few more in the appendix – by my count, 2). Do all experiments take a uniform target distribution? If so, this also seems restrictive.

**Methods And Evaluation Criteria:**

The authors compare a variety of MCMC algorithms against their new “history-driven target” (HDT) counterparts, and consistently find faster convergence to a uniform target distribution. They consider facebook (N=4039) and p2p-Gnutella04 (N=10876), with a few more graphs in the appendix. They also compare to SRRW for fixed computational budget. Whilst this convincingly backs up their core claims, I wonder if the authors could have tried some more ambitious applications of graph-based MCMC to show real-world advantages of HDT over existing methods – i.e. faster convergence to the target distribution actually unlocking better performance in things directly relevant to ML practitioners.

**Other Comments Or Suggestions:**

N/A

**Other Strengths And Weaknesses:**

**Strengths**:
- Clear presentation of a novel idea (history-driven targets), capable of enhancing a range of previous graph-based MCMC samplers using information about walker history
- Impressive theoretical contributions
- Central claims of more efficient graph sampling are backed up by Figs 2 and 3
- LRU caching scheme seems reasonable to further reduce computational cost

**Weaknesses**:
- Limited number of graphs considered, and limited evidence of benefits for downstream ML tasks (though I don’t doubt that they exist)

**Questions For Authors:**

- Can you demonstrate downstream benefits from improved efficiency of graph sampling in ML algorithms?
- How do the gains provided by your scheme depend on the properties of the graph? What properties of the graph govern the size of the gains? (I appreciate that this is a big one…) Also, how do the gains depend on the target measure?

**Relation To Broader Scientific Literature:**

Improving MCMC graph sampling algorithms to encourage efficient exploration is a storied line of work – e.g. Metropolis Hastings with delayed rejection reduces repeated rejections (Green & Mira, 2001), and multiple-try Metropolis proposes multiple candidates at each step and picks one based on their weights (Liu et al., 2000; Pandolfi et al., 2010). The authors’ method can be used to enhance each of these variants by introducing a history-dependent target distribution, as demonstrated in Fig. 2. It provides a computationally cheaper alternative to SRRWs, introduced by Doshi et al. in 2023.

**Theoretical Claims:**

The authors prove that their HDT framework converges almost surely to the original target distribution and achieves variance reduction compared to the base sampler in the CLT. They show further variance reduction relative to SRRW that depends on the average neighborhood size. I did not check the proofs in detail, but all theoretical proofs seem reasonable.

---

> ### Author Rebuttal · Authors · 2025-03-30
>
> ## Q1. Downstream benefits in ML algorithms
> Our HDT framework enables more efficient sampling of large or complex graphs for arbitrary target measures, reducing estimator variance and cost. Key impacts include:
> 1. **Estimating Global Properties of Large-Scale Graphs:** In social networks, e-commerce systems, recommender systems and other domains, practitioners need to estimate graph size, node degree distribution, vertex label distribution [Xie et al. 2023]. These applications include detecting bot populations, identifying high-value user segments or scarce features [Nakajima et al. 2023], and gradient of local loss for token algorithms in distributed optimizations [Sun et al. 2018, Even 2023], especially with limited graph access. However, a standard random walk can be slow to discover underrepresented regions or rare features, forcing more samples to achieve acceptable levels of accuracy.
> 2. **Sampling from Large-Scale Graphs:** In statistical physics (e.g., Ising or Potts models) [Grathwohl et al. 2021, Zhang et al. 2022], machine learning (e.g., Restricted Boltzmann machines / energy-based models for image generation, reconstruction or partition, Bayesian variable selection for high-dimensional regression/classification) [Hinton 2010, Zanella 2020, Rhodes et al. 2022, Chang et al. 2022, Sun et al. 2023], and econometrics (e.g., MCMC-MLE method for exponential random graph models) [Bhamidi et al. 2011, Byshkin et al. 2016], the general discrete state space can always be translated into a graph where each node represents a possible configuration and edges link configurations that differ by the Hamming distance. Efficient exploration yields quicker, more accurate system representation and inference.
>
> HDT-MCMC strategically modifies the target distribution at minimal extra cost compared to baseline MCMC and SRRW, achieving equivalent accuracy with fewer samples and computations. While fully deployment in exponentially large configuration state spaces present ongoing memory challenges beyond this paper's scope, our method offers versatile and simple implementation alongside significant efficiency improvements for sampling on general graphs. We plan to detail these applications in a revised appendix if our paper is accepted.
>
> [*References not included in our manuscript are listed below for this response.*]
>
> G. Hinton. "A practical guide to training restricted Boltzmann machines." Momentum, 2010.
>
> S. Bhamidi et al. "Mixing time of exponential random graphs." Ann. Appl. Probab., 2011.
>
> M. Byshkin et al. "Auxiliary parameter MCMC for exponential random graph models." J. Stat. Phys., 2016.
>
> T. Sun et al. "On markov chain gradient descent." NeurIPS, 2018.
>
> H. Xie et al. "A bootstrapping approach to optimize random walk based statistical estimation over graphs." IEEE TKDE, 2023.
>
> K. Nakajima et al. "Random walk sampling in social networks involving private nodes." ACM TKDD, 2023.
>
> M. Even. "Stochastic gradient descent under markovian sampling schemes." ICML, 2023.
>
> ---
>
> ## Q2. Impact of graph properties on the gain of HDT-MCMC
> The gains of HDT-MCMC relative to baseline MCMC and SRRW are addressed as follows:
> 1. **HDT-MCMC v.s. Baseline MCMC:** Our Theorem 3.3 shows HDT-MCMC reduces sampling variance by a factor of $1/(2α+1)$ compared to baseline MCMC:
> $$
> V^{HDT} (α)=\\frac{1}{2α+1} V^{base}.
> $$
> $V^{base}$ is defined via transition kernel’s spectrum in Eq. (4) in Line 215 and is also related to spectral gap and mean hitting time [Aldous et al. 2014, Remark after Theorem 2.17 & Eq. (11.4)], where graph with higher connectivity usually ensures smaller spectral gap and mean hitting time. However, HDT-MCMC *uniformly* reduces variance by the $1/(2α+1)$ factor. Corollary 3.4 adds that a more efficient baseline *amplifies* this gain: the better the baseline, the more HDT lowers the variance, regardless of the graph or target measure.
> 2. **HDT-MCMC v.s. SRRW:** Our Lemma 3.6 highlights HDT-MCMC’s cost-based variance reduces by a factor of $\\frac{2}{\\mathbb{E}\_{i∼μ} [|\\bar{\\mathcal{N}}(i)|]}$ compared to SRRW. The average degree $\\mathbb{E}\_{i∼μ} [|\\bar{\\mathcal{N}}(i)|]$ (including self-loops) depends on graph density and the target measure $\mu$. Even for a sparse graph with an average degree of $3$, the factor is smaller than $0.5$, translated into over 50% variance reduction compared to SRRW under limited budgets. Gains are more pronounced in denser graphs (higher average degree under $\mu$), making HDT particularly effective for large, high-degree networks, as shown numerically in Figure 3 and around Lines 418 - 423.
>
> D. Aldous et al. "Reversible Markov chains and random walks on graphs." Unfinished monograph, 2014.
>
> ---
> ## Additional experiments on non-uniform target
> While we use the uniform target in our paper, we've added experiments targeting degree distribution on more graphs to address this. Due to space limits, these results are in the PDF with a link in the response to reviewer mrvq.

---

> > ### Comment · Reviewer_nUqH · 2025-04-06
> >
> > Thanks for the detailed response. I'll be happy to upgrade my score.

---

### Official Review · Reviewer_mrvq · 2025-03-12

**Overall Recommendation:** 4

**Summary:**

This paper introduces the History-Driven Target (HDT) framework, a novel approach to improving Markov Chain Monte Carlo (MCMC) sampling on discrete state spaces, such as general undirected graphs. The HDT framework replaces the original target distribution with a history-dependent target distribution $ \boldsymbol{\pi}[\mathbf{x}] $, which adapts dynamically based on past visit frequencies. This design achieves compatibility with both reversible and non-reversible MCMC samplers, reduces computational overhead compared to Self-Repellent Random Walks (SRRW), and provides unbiased sampling with near-zero variance performance. The authors also propose a memory-efficient Least Recently Used (LRU) cache scheme to address scalability challenges in large graphs. Extensive experiments demonstrate consistent performance gains across various graph types and MCMC methods.

**Claims And Evidence:**

The paper makes several clear claims:
- HDT is compatible with both reversible and non-reversible MCMC samplers.
- HDT achieves $ O(1/\alpha) $ variance reduction relative to baseline MCMC samplers.
- HDT avoids SRRW's computational overhead by decoupling self-repellency from the transition kernel.
- The LRU cache scheme enables HDT-MCMC to scale efficiently to large graphs.

**Essential References Not Discussed:**

The authors provide a comprehensive review of related works but could include additional references on stochastic approximation techniques relevant to their framework.

**Experimental Designs Or Analyses:**

The experimental designs are valid:
- Experiments test HDT-MCMC across reversible and non-reversible chains on real-world graphs.
- Results consistently show improvements in TVD and NRMSE compared to baseline methods.
- Simulations investigate the effect of key parameters like $ \alpha $, demonstrating its impact on convergence speed.

**Methods And Evaluation Criteria:**

The methods proposed are appropriate for the problem at hand:
- The HDT framework is rigorously designed to preserve lightweight implementation while achieving self-repellency through adaptive target distributions.
- Evaluation metrics include TVD and RMSE, which effectively measure convergence speed and estimator accuracy.
- Experiments span diverse graph types and include comparisons between HDT-enhanced algorithms and baseline MCMC methods.

**Other Comments Or Suggestions:**

None.

**Other Strengths And Weaknesses:**

**Strengths**

- The HDT framework introduces a novel way to achieve self-repellency without modifying transition kernels.
- The contributions are well-explained, with detailed experiments and proofs provided in the main text and appendix.

**Weaknesses**
- The paper does not explore how different initializations affect empirical results or robustness.
- No confidence intervals or sensitivity analyses are provided for metrics like TVD or NRMSE under varying conditions.

**Questions For Authors:**

How do different initialization strategies (e.g., initial sample state or fake visit counts) affect convergence speed or variance reduction? Have you tested robustness under varying initialization? Maybe providing uncertainty quantification for TVD or NRMSE metrics across multiple runs would be beneficial.

**Relation To Broader Scientific Literature:**

The paper builds upon prior work in MCMC sampling (e.g., Metropolis-Hastings, SRRW) while addressing their limitations:
- SRRW introduced self-repellency but suffered from high computational costs and reliance on time-reversibility.
- Gradient-based methods are efficient but limited to structured spaces.

**Theoretical Claims:**

The proofs of convergence to $ \boldsymbol{\mu} $, variance reduction, and global asymptotic stability are detailed and mathematically rigorous.

  - Lemma 3.1 ensures that $ \boldsymbol{\pi}[\mathbf{x}] $ satisfies scale invariance, local dependence, fixed-point properties, and history dependence.
  - Theorems 3.3 and 3.5 provide ergodicity and cost-based CLT results that align with stochastic approximation theory.

I did not rigorously check the theoretical claims, but it generally looks reasonable to me.

---

> ### Author Rebuttal · Authors · 2025-03-30
>
> Thank you for your constructive and practical feedback. We agree that demonstrating robustness to initialization and quantifying uncertainty in our results will enhance the paper. Below is the setting of additional experiments. Due to space constraints, we defer the simulation results on more real-world graphs under TVD and NRMSE metrics in the anonymous linked PDF:
>
> https://www.dropbox.com/scl/fi/ujmodl9e8721nef5dx2k5/ICML_2025_Rebuttal_Additional_Simulations.pdf?rlkey=80cp2rhdvf59cvjukrtrcz5km&st=b96ipxpx&dl=0
>
> ---
>
> ## Robustness of HDT-MCMC to different initialization strategies and uncertainty quantification.
> We run $1000$ independent trials for different MCMC methods on the wikiVote graph with uniform target and show the TVD $\mathbb{E}[||x_n-\mu||_1]$ at $n=3000$ steps with 95% confidence intervals:
> 1. **Varying the Initial State $X_0$:**
>    Besides random selection, we started from nodes in different regions of the graph:
>     - Low-degree group (nodes with below-average degree)
>     - High-degree group (nodes with above-average degree)
>
>     ---
>     **Table 1.** Mean (std error) TVD with different $X_0$ initializations. *All std error values are in units of* \\(10^{-4}\\).
>
>     | **Method**| **Low-Deg Group**| **High-Deg Group**|
>     |--------------|-----------------|-----------------|
>     | MHRW         | 0.573 (7.316)    | 0.573 (7.290)    |
>     | HDT-MHRW     | 0.417 (3.785)    | 0.417 (3.804)    |
>     | MTM          | 0.533 (6.818)    | 0.533 (7.076)    |
>     | HDT-MTM      | 0.331 (5.417)    | 0.330 (4.154)    |
>     | MHDA         | 0.544 (7.419)    | 0.543 (7.185)    |
>     | HDT-MHDA     | 0.395 (3.564)    | 0.395 (3.569)    |
>     ---
>
> 2. **Varying the Fake Visit Count $x_0$:**
> We use three scenarios:
>     - 'Deg' scenario: Fake visit counts set by node degrees, heavily favoring high-degree nodes.
>     - 'Non-unif' scenario: Fake visit counts randomly drawn from a Dirichlet distribution with default concentration parameter of $0.5$.
>     - 'Unif' scenario: Same initial count for all nodes.
>
>     In this part, we start with $X_0$ uniformly at random. Note that baseline methods (MHRW, MTM, MHDA) do not depend on fake visit counts $x_0$, so they give identical TVD values in all three scenarios.
>
>     ---
>     **Table 2.** Mean (std error) TVD with different $x_0$ initializations. *All std error values are in units of* \\(10^{-4}\\).
>
>     | **Method**   | **Deg**       | **Non-unif**   | **Unif**       |
>     |--------------|---------------|---------------|----------------|
>     | MHRW         | 0.573 (7.085)  | 0.573 (7.085)  | 0.573 (7.085)   |
>     | HDT-MHRW     | 0.416 (3.767)  | 0.416 (3.877)  | 0.416 (3.810)   |
>     | MTM          | 0.532 (7.092)  | 0.532 (7.092)  | 0.532 (7.092)   |
>     | HDT-MTM      | 0.325 (4.301)  | 0.332 (4.200)  | 0.330 (4.130)   |
>     | MHDA         | 0.543 (7.219)  | 0.543 (7.219)  | 0.543 (7.219)   |
>     | HDT-MHDA     | 0.395 (3.605)  | 0.395 (3.657)  | 0.396 (3.587)   |
>     ---
>
> In both Tables 1 and 2, HDT-MCMC shows robust convergence and consistent variance reduction relative to the baseline. We observe that the initialization effect quickly vanishes during the burn-in period (one third of the total samples) so they do not affect the performance of HDT-MCMC. Even when the initial visitation $x_0$ is heavily skewed (such as degree-based distribution where weight goes mostly to the node with largest degree), the HDT forces rapid exploration of under-sampled regions.
>
> In most cases, the standard error for our HDT-MCMC framework is smaller than baseline algorithms, indicating the consistency of HDT-MCMC across repeated runs. This shows that every sample path generated by our HDT framework performs consistently. These results suggest that the HDT $π[x]$ contributes to stable sampling trajectories that rapidly converge to the target distribution $μ$. We conjecture that this consistent performance of HDT-MCMC is due to the correctness behavior from our design of history-aware scheme in every trajectory to drive $x_n$ closer to the target distribution $μ$.
>
> We will include detailed simulation setups and plots in the revised paper to illustrate these findings if our paper is accepted.

---

> > ### Comment · Reviewer_mrvq · 2025-04-07
> >
> > I submitted an official comment, but it was not visible to the authors. So I am reposting it here:
> >
> > Thanks for the authors' response. I will maintain my current score.

---

### Official Review · Reviewer_Fmtv · 2025-03-15

**Overall Recommendation:** 5

**Summary:**

- Propose a history-driven target (HDT) framework, extending the self-repellent random walk (SRRW) framework on general graphs [Doshi et al., 2023] to allow for construction from a non-reversible sampler, whereas the original framework in [Doshi et al., 2023] required a reversible sampler.

- On the practical side, HDT exhibits local dependence, meaning that each particle update only requires local information rather than accessing information from all its neighbors. This significantly reduces per-step computational cost compared to SRRW, which is particularly important for walking on large, dense graphs. The authors provide both theoretical analysis and empirical evidence demonstrating the computational efficiency of HDT.

- The authors also introduce an efficient implementation of HDT with improved memory scalability over SRRW, making it more practical for use on large-scale graphs.

**Claims And Evidence:**

- The main claims are:
1. The HDT sampler (alg 1) converges to the right target.
2. HDT allows for both non-reversible and reversible sampler.
3. better performance/ cost ratio than the SRRW.
- All claims are well supported by both theoretical investigations and empirical results.

**Essential References Not Discussed:**

n/a

**Experimental Designs Or Analyses:**

good

**Methods And Evaluation Criteria:**

they make sense.

**Other Comments Or Suggestions:**

n/a

**Other Strengths And Weaknesses:**

Strength:

- Well-written and self-contained: The paper is exceptionally well-written, with a concise yet comprehensive review of SSRW. The exposition is clear, making it accessible to a broad audience.

- Clarity in problem formulation: The authors clearly outline the challenges this work aims to address and provide careful explanations of the proposed methodological components. The discussion in Lines 212–245 is particularly well-structured and insightful. Similarly, each theoretical result is accompanied by a concise discussion of its implications, which greatly enhances readability.

- Strong theoretical foundation: The theoretical results are solid, with well-organized proofs. Even the appendix is a pleasure to read, demonstrating the rigor and care put into the analysis.

- Versatility and practical impact: The proposed algorithm is highly versatile and simple to implement. Algorithm 1 is not only elegant but also provably enhances vanilla MCMC performance on graphs at no additional cost. Given its efficiency and general applicability, I believe this approach will set a new standard for graph-based sampling and become the default algorithm in this domain.

I have a VERY positive opinion on this work; it is not very common to see submissions like this with good writing, careful theoretical analysis, simple algorithm, and superior performance.

**Questions For Authors:**

see previous comments.

**Relation To Broader Scientific Literature:**

It's relavent to the literature of building efficient MCMC on general graphs. Particularly, it extends the recent advances on SRRW [Doshi et.al. 23]. Many proof techniques are from the CLT type analysis for stochastic aproximations Delyon (2000) and Fort (2015).

The authors discussed relevant literatures whenever needed.

**Theoretical Claims:**

I checked all the proofs (with particular  focus on the proof of Theorem 3.3); I believe they are all correct. I do have a few minor questions though, but I do not think there are any correctness issues.

Questions:

- Line 654: It's not trivial to me why $g(c1, c2, x_i, \mu_i)$ has to be independent to $x_i, \mu_i$. Could you please provide a proof by contradiction that if $g$ indeed depends on $x, \mu$, things won't work?
- Minor typo: in Eq(24) the step size should be $\gamma_n$ instead of $1/(n+1)$?
- About non-reversability: As the author mentioned in Line 1030 - line 1032, the proof in E.1. and E.2. does not explicitly require the reversability of the Markov chain that drives $X_n$. To To my awareness, the proof only requires that $P[x]$ preserves $\pi[x]$, which is true by construction. So why does one need extra care about the non-reversable case? I do understand that some non-reversible Markov chain, by design, operates on a augmented state space, but they typically still maintain the right stationary distribution in the subspace that we care about (I'd love to be proven wrong here).  And to my knowledge, a Markov process is ergodic (not geometric ergodic) if it obeys a unique stationary distribution; unless the uniqueness here is violated in some non-reversability case, I don't see why the original proof doesn't apply.

Good things about the theory: I've learned many new techniques when reading the proof of this work (though I'm sure it's due to my lack of knowledge in recent advances of MCMC on graphs/general discrete spaces). I find the following things particularly interesting:
- Lemma 3.1: using ODE stability analysis to show that the Markov process exhibits a unique stationary distribution.
- Theorem E.1: using general theory of SGD to show ergodicity and CLT of the sampler. Though this has been used in previous works as discussed by the authors.
- The cost aware CLT is also new to me.

---

> ### Author Rebuttal · Authors · 2025-03-30
>
> Thank you for your thoughtful and detailed review. We appreciate your positive remarks on our paper’s theoretical depth and are grateful for the questions you raised. Below, we address each of your points in turn.
>
> ---
>
> ## Response to Q1
>
> We agree that this point warrants further explanation. Here is a streamlined proof by contradiction:
> 1. Assume that $g(c_1,c_2,x,\\mu)$ depends on $x, \\mu$.
> 2. According to Line 650, for any coordinate $i \\in [N]$
> $$
> f(c_1 x_i,c_2 μ_i )=g(c_1,c_2,x,μ)f(x_i,μ_i),
> $$
> where the function $f :\\mathbb{R}\_{>0}×\\mathbb{R}\_{>0}→\\mathbb{R}\_{>0}$.
> 3. Consider two pairs $(x,μ)$ and $(\\hat x,\\hat μ)$ whose $i$-th coordinate are identical (i.e., $x_i=\\hat x_i,μ_i=\\hat μ_i$). Then, $f(c_1 x_i,c_2 μ_i )=f(c_1 \\hat x_i,c_2 \\hat μ_i )$ and $f(x_i,μ_i )=f(\\hat x_i,\\hat μ_i )$. By the equation in step 2, it follows that
> $$
> g(c_1,c_2,x,μ)=g(c_1,c_2,\\hat x, \\hatμ).
> $$
> 4. Similarly, construct another pair ($\\tilde x,\\tilde μ$) (i.e., $\\tilde x_j=\\hat x_j,\\tilde μ_j=\\hat μ_j$ for $j$-th coordinate) such that $g(c_1,c_2,\\tilde x, \\tilde μ)=g(c_1,c_2,\\hat x,\\hat μ)$. Then, $(\\hat x, \\hat μ)$ serves as a ‘bridge’ to $(x,μ)$ and $(\\tilde x,\\tilde μ)$, leading to
> $$
> g(c_1,c_2,x,μ)=g(c_1,c_2,\\tilde x,\\tilde μ).
> $$
> For a given pair $(x,μ)$, we can always construct a distinct pair $(\\tilde x,\\tilde μ)$ such that $g(c_1,c_2,x,μ)=g(c_1,c_2,\\tilde x, \\tilde μ)$, indicating that function $g$ must be identical for any $(x,μ)$. Hence $g$ cannot actually depend on $x$ or $μ$; it must be a function of $(c_1,c_2)$ alone.
>
> We will revise Lines 653–655 in the manuscript with this explanation to clarify why $g$ must be independent of $(x,μ)$.
>
> ---
>
> ## Response to Q2
>
> Thank you for catching this. We will correct the mentioned typo and any others in the revision.
>
> ---
>
> ## Response to Q3
>
> This is an excellent question. You are correct that Appendix E.1 and E.2 do not assume reversibility, allowing non-reversible chains to operate directly on the state space $\\mathcal{X}$ of interest. Let $\\mathcal{Y}$ be the auxiliary state space. Below, we explain why analyzing non-reversible chains on the augmented state space $\\mathcal{Z}≜\\mathcal{X}×\\mathcal{Y}$ is more than $P[x]$ preserving $π[x]$, and discuss the technical challenges in directly applying the stochastic approximation (SA) framework, as well as how Appendix E.3 addresses them.
>
> 1. **State Space Inconsistency in SA Analysis**
> The SA iteration
> $$
> x_{n+1}=x_n+γ_{n+1} H(x_n,X_{n+1})≡x_n+\\frac{1}{n+1}(δ_{X_{n+1}}-x_n)
> $$
> requires $\\{X_n\\}$ to be *controlled Markovian dynamics*. That is, $X_{n+1}$ is drawn from a transition kernel $P_{(X_n,⋅)} [x_n]$ parameterized by empirical measure $x_n$ (in Line 891). If the chain instead operates on an augmented space $\\mathcal{Z}$, then $\\{X_n\\}$ *in isolation* cannot be viewed as a controlled Markov chain, since $X_{n+1}$ depends not only on $X_n, x_n$, but also on the auxiliary state $Y_n\\in \\mathcal{Y}$.
> More concretely, mixing these spaces (using a transition matrix on $\\mathcal{Z}$ but a function $H(x,⋅)$ only defined on $\\mathcal{X}$) causes a dimensional mismatch in the SA analysis. For example, in Eq. (32) around Line 952, the solution
> $$
> m_x (i)=∑_{j∈\\mathcal{Z}} (I-P[x]+1π[x]^T)_{ij}^{-1} (H(x,j)-h(x))
> $$
> to the Poisson equation is not well-defined because $P[x]$ is specified on $\\mathcal{Z}$, whereas $H(x,⋅)$ is only on $\\mathcal{X}$. From the viewpoint of SA iteration, a direct redefinition of $H$ via
> $$
> H:(0,1)^{|\\mathcal{X}|}×\\mathcal{Z}→\\mathbb{R}^{|\\mathcal{X}|}:(x,Z)→δ_Z-x
> $$
> does not resolve the core mismatch because $δ_Z$ (a canonical vector in $\\mathcal{Z}$) does not align with the dimension of $x$ (an empirical measure in $\\mathcal{X}$).
>
> 2. **Dimension and Memory Overhead**
> A natural solution to the above mismatch is to record visit frequencies for the augmented space $\\mathcal{Z}$, thereby resolving dimensional mismatch but *inflating* the dimension of $x$. For instance, this would increase the dimension of $x$ from $|\\mathcal{X}|$ to $|\\mathcal{X}|^2$ for MHDA (Lee et al., 2012) and to $2|\\mathcal{X}|$ for the 2-cycle Markov chain (Maire et al., 2014). This is especially problematic for large state space $\\mathcal{X}$.
>
> **Our “Trick” in Appendix E.3**
> We address these issues by replacing $H$ with
> $$
> ϕ: (0,1)^{|\\mathcal{X}|}×\\mathcal{Z}→\\mathbb{R}^{|\\mathcal{X}|}:(x,(X,Y))→δ_X-x
> $$
> so that the augmented state $Z∈\\mathcal{Z}$ is 'accepted' by $\phi$, but we *only* store visit frequencies for $\\mathcal{X}$ (i.e., we project $\\mathcal{Z}$ onto $\\mathcal{X}$). This reconciles the SA analysis with the non-reversible Markov chain on $\\mathcal{Z}$ (via the function $\phi$) without inflating the dimension of $x$.
>
> We will revise Line 1043 in the manuscript to clearly emphasize how this approach extends our SA analysis to non-reversible chains on augmented state spaces, mitigating both dimensional mismatch and memory concerns.

---

> > ### Comment · Reviewer_Fmtv · 2025-04-06
> >
> > Thank you for the detailed response. I'd love to maintain my recommendation for this work.

---

### Decision · Program_Chairs · 2025-05-01

**Decision:**

Accept (oral)

**Comment:**

**Summary.**

The topic of this work is MCMC (Markov Chain Monte Carlo) on discrete spaces. Specifically, the paper introduces the so-called **HDT (History-Driven Target)** framework for improving existing MCMC sampling algorithms over discrete spaces.

The method shares strong conceptual similarities with the previously introduced **SRRW** (Self-Repellent Random Walks) [Doshi et al., 2023] framework, in as much as the method is **adaptive to the empirical distribution**: less visited states tend to gain increased probabilities of future visits.

However, the key conceptual difference is that one modifies the target distribution based on the current empirical distribution, instead of modifying the transition kernel (SRRW).

As a secondary contribution, the authors propose a cache scheme for reducing memory requirements.

**Strengths.**

* The method achieves a variance reduction compared to the base sampler which is at least similar to SRRW and even more pronounced on denser graphs.
* While SRRW could only be applied to reversible MCMC algorithms, the newly proposed method can also be applied to non-reversible MCMC algorithms.
* The new method is less computationally expensive than SRRW.
* The quality of the writing is very high. Clarity of the paper has been praised by the referees.
* Both theoretical and empirical justification for the claims.

**Weaknesses.**

* The experimental value was considered to lack ambition.

**Discussion and reviewer consensus.**

* There is a consensus among the reviewers that the paper should be accepted.
* Reviewers are enthusiastic about the paper; it has received a lot of praises.
* There was an original concern about initialization, but that was rebutted.